



# Contributions of meteorology and anthropogenic emissions to the trends in winter PM2.5 in eastern China 2013−2018

Yanxing Wu[1], Run Liu[1,2*], Yanzi Li[3], Junjie Dong[1], Zhijiong Huang[1,2], Junyu Zheng[1,2] and Shaw Chen Liu[1,2*]

[1]Institute for Environmental and Climate Research, Jinan University, Guangzhou, 511443, China
[2]Guangdong-Hongkong-Macau Joint Laboratory of Collaborative Innovation for Environmental Quality, Guangzhou, 511443, China
[3]Guangzhou Huayue Technology Co., Ltd., Guangzhou, 510630, China

*Correspondence to*: Run Liu (liurun@jnu.edu.cn), Shaw Chen Liu (shawliu@jnu.edu.cn)

**Abstract.** Multiple linear regression (MLR) models are used to assess the contributions of meteorology/climate and anthropogenic emission control to linear trends of PM2.5 concentration during the period 2013–2018 in three regions in eastern China, namely Beijing-Tianjin-Hebei (BTH), Yangtze River Delta (YRD), and Pearl River Delta (PRD). We find that quantitative contributions to the linear trend of PM2.5 derived based on MLR results alone are not credible because a good correlation in the MLR analysis does not imply any causal relationship, let alone a quantitative relationship. As an alternative, we propose that the correlation coefficient should be interpreted as the maximum possible contribution of the independent variable to the dependent variable, and the residual should be interpreted as the minimum contribution of all other independent variables. Under the new interpretation, the MLR results become self-consistent. We also find that the results of a short-term (2013–2018) analysis are significantly different from those of a long-term (1985–2018) analysis for the period 2013–2018 they overlap, indicating that MLR results depend critically on the length of time analyzed. The long-term analysis renders a more precise assessment, because of additional constraints provided by the long-term data. We therefore suggest that the best estimates of the contributions of emissions and non-emission (including meteorology/climate) to the linear trend in PM2.5 during 2013–2018 are those from the long-term analyses: i.e., emission <51% and non-emission >49% for BTH, emission <44% and non-emission >56% for YRD, emission <88% and non-emission >12% for PRD.

## 1 Introduction

PM2.5 (particulate matter with an aerodynamic diameter less than 2.5 µm) pollution is a severe problem in China that affects human health (Kan et al., 2007; Wang and Mauzerall, 2006; Xu et al., 2013; Cohen et al., 2017), visibility (Han et al., 2014; Zhang et al., 2012; Zhang et al., 2014), the acid deposition problem (Yim et al., 2019; Zhang et al., 2016) and the climate systems (Albrecht, 1989; Carslaw et al., 2010; Kok et al., 2018). Recent observations from the China National





Environmental Monitoring Center have shown a 30%–50% decrease in annual mean PM$_{2.5}$ concentration in China during 2013–2018 (Zhai et al., 2019).

These remarkable decreases in the PM$_{2.5}$ concentrations have been mostly attributed to emission control of PM$_{2.5}$ and its precursors in several recent studies (e.g., Chen et al., 2019; Gong et al., 2021; Zhai et al., 2019). Using various statistical models, these studies concluded that the control of anthropogenic emissions accounted for 81% to 103% of the reductions of

PM$_{2.5}$ in eastern China, suggesting that emissions reductions are crucial to the improvement of air quality in 2013–2018 (Chen et al., 2019; Gong et al., 2021; Zhai et al., 2019). However, Dang and Liao (2019) conducted an investigation with a global 3-D chemical transport model and revealed that transport is the most important process for the occurrence of severe winter haze days in Beijing–Tianjin–Hebei (BTH) in 2013–2017. Although Dang and Liao (2019) dealt with the severe winter haze rather than the mean haze in BTH, their results suggest that meteorological conditions may exert a critical

impact on the reduction in PM$_{2.5}$ in eastern China.

In this study, we use multiple linear regression (MLR) models to investigate the relative contributions of emissions control and climate/meteorology to linear trends in winter PM$_{2.5}$ concentration in three major polluted regions in eastern China, namely BTH, Yangtze River Delta (YRD), and Pearl River Delta (PRD). The results can provide important insight for better designing successive clean-air plans to further mitigate PM$_{2.5}$ pollution in China. The rest of the paper is structured as

follows: The data and methods employed are introduced in Section 2, Section 3 presents the major results and discussions, and Section 4 presents a summary the conclusions.

## 2 Data and methodology

### 2.1 Data

Winter visibility data in 1973–2019 are obtained from Global Summary of Day (GSOD) provided by the National Climatic

Data Center (NCDC) (https://gis.ncdc.noaa.gov/maps/ncei/cdo/daily, last access: 10 March 2022). Surface PM$_{2.5}$ measurements in 2013–2019 are taken from China National Environment Monitoring Center (CNEMC, http://www.cnemc.cn/, last access: 10 March 2022). The PM$_{2.5}$ concentrations are measured by the micro-oscillating balance method and/or the β-absorption method (MEE, 2012; Zhang and Cao, 2015). Arctic sea ice (ASI) data are taken from the Hadley Centre Sea Ice data set (https:// https://www.metoffice.gov.uk/hadobs/hadisst/, last access: 10 March 2022).

PM emission inventory of PM$_{10}$, PM$_{2.5}$, SO$_2$, NH$_3$, NOx, black carbon, and organic carbon in this study is obtained from Peking University (PKU, 1960–2014, 0.1º×0.1º, monthly), which includes the fuel consumption and emissions of greenhouse gases and air pollutants from all combustion sources (http://inventory.pku.edu.cn/, last access: 10 November 2021). Multi-resolution Emission Inventory for China (MEIC, version 1.3, 2010–2017, 0.25º×0.25º, monthly, http://www.meicmodel.org, last access: 10 November 2021) provided by Tsinghua University and PRD Emission Inventory

(PRD-EI, 2006–2019, 3 º×3º, monthly) from Huang et al. (2021) and Zhong et al. (2018) are also used. Due to the



discontinuity of these three inventories, we calculate the scaling factor of each pollutant based on the overlapping period to get a winter inventory from 1985–2018 (Figure 1) in PRD as follows:

$$\text{scaling factor}_{Ei} = \frac{\sum_{2013}^{2006} Ei_{j,PKU}}{\sum_{2013}^{2006} Ei_{j,PRD-EI}}$$

$$E_i = \begin{cases} Ei_{j,PKU} \ (j=1985, 1986, \dots 2013) \\ \text{scale factor}_i \times Ei_{j,PRD-EI} \ (j=2014, 2015, 2016, 2017, 2018) \end{cases}$$

where $E_i$ is emission of species i, the subscripts PKU and PRD-EI denote the PKU inventory and the PRD inventory of Huang et al. (2021) and Zhong et al. (2018), respectively, and j denotes year.

The corresponding formulas for BTH and YRD are:

$$\text{scaling factor}_{Ei} = \frac{\sum_{2013}^{2010} Ei_{j,PKU}}{\sum_{2013}^{2010} Ei_{j,MEIC}}$$

$$E_i = \begin{cases} Ei_{j,PKU} \ (j=1985, 1986, \dots 2013) \\ \text{scale factor}_i \times Ei_{j,MEIC} \ (j=2014, 2015, 2016) \end{cases}$$

where the subscript MEIC denotes the MEIC inventory.

Given that emission is not expected to change significantly in one or two years, the 2016 winter emission ratio of BTH to PRD is multiplied by the 2017 and 2018 winter emission of PRD to obtain the winter emission of BTH in 2017 and 2018. The same is true for 2017 and 2018 winter emission of YRD. The annual emission inventories for BTH, YRD and PRD from 1985 to 2018 are shown in Figure 1.

**2.2 Nonlinear exponential fitting**

Since direct observed data of $PM_{2.5}$ are not available before 2013, we employ nonlinear exponential fitting to retrieve $PM_{2.5}$ concentrations in BTH, YRD and PRD from visibility that has long-term and complete record. Because relative humidity (RH) affects strongly the relationship between $PM_{2.5}$ concentration and visibility (Fu et al., 2016; Liu et al., 2017; L. Zhang et al., 2015; Q. Zhang et al., 2015), we evaluate the relationship for different RH intervals in each region as shown in the

Supplementary Materials. The $r^2$ of average fitting is greater than 0.5, some as high as 0.77, significant at 99% confidence level, indicating that the fitting performance is acceptable (Figures S1–S3). The retrieved $PM_{2.5}$ has a significant negative correlation with visibility as expected, and is consistent with the trend of observed $PM_{2.5}$ in recent years (Figure S4). The exponential fitting model captures the long-term trend and the interannual variation of $PM_{2.5}$ well (Figure 1).

A quick inspection of the $PM_{2.5}$ and emission lines in Figure 1 reveals that an expected good regression is only possible for

PRD where the emission line matches well with the $PM_{2.5}$ line in general as both have a broad maximum near 2005–2010 (Figure 1c). In BTH a good regression is made difficult due to a key mismatch characterized by two broad maxima (1985–2000, 2000–2013) in the emission line and a sharp drop after 2013, while the $PM_{2.5}$ line had a shallow depression during the period 1993–2012 followed by a large bulge peaked in 2013 (named hereafter as bulge-2013) and lasted until 2016 (Figure



1a). This mismatch also existed in YRD as the emission line crossed over the PM$_{2.5}$ line in opposite directions near 2012,
albeit the bulge was not as large and a depression could be barely seen in 2003–2012 (Figure 1b).

Mao et al. (2019) have analyzed extensively the depression 1999–2012 (the deeper part of the 1993–2012 depression) and bulge-2013. They found that the depression and bulge-2013 were primarily caused by a combination of climatic oscillations which include negative phases of the Pacific Decadal Oscillation, Arctic Oscillation, El Niño-Southern Oscillation and global temperature, in addition to positive phases of East Asian Winter Monsoon and ASI. It is clear that the presence of
bulge-2013 can have a big impact on any study on the contributions of meteorology/climate and anthropogenic emission control to the linear trends in PM$_{2.5}$, especially for a short-term study such as the period 2013–2018 that overlooks the cause of the bulge. This point will be elaborated in Sections 3.3 and 3.4.

## 3 Results and discussions

### 3.1 Multiple linear regression studies

Zhai et al. (2019) constructed a stepwise multiple linear regression (MLR) model to quantify the meteorological contribution to the PM$_{2.5}$ trends. The MLR model correlates the 10-day PM$_{2.5}$ anomalies to wind speed, precipitation, RH, temperature, and 850 hPa meridional wind velocity. The meteorology-corrected PM$_{2.5}$ trends obtained by removing meteorological contribution viewed as being driven by trends in anthropogenic emissions. They quantified that the mean PM$_{2.5}$ decrease in BTH, YRD and PRD from 2013 to 2018 is 86%, 103% and 81% in the meteorology-corrected data respectively, meaning
that 14%, -3% and 19% of the PM$_{2.5}$ decrease in the original data is attributable to meteorology (Table 1).

Chen et al. (2019) employed Kolmogorov–Zurbenko (KZ) filter to produce an adjusted long-term time series of PM$_{2.5}$ concentrations in Beijing from 2013 to 2017 by removing interannual and seasonal variations in meteorological conditions. They applied MLR models between PM$_{2.5}$ and wind speed, RH, temperature and solar radiation to remove the influence of meteorological conditions and then quantified that the contribution of emissions to adjusted PM$_{2.5}$ was 81%, while the
contribution of meteorology was 19% (Table 1).

To compare with these two studies, we carry out an MLR analysis on the emission and observed concentrations of PM$_{2.5}$, of which the results will be denoted hereafter as MLR-EMIS. As shown in Figure 2, the observed PM$_{2.5}$ decreased remarkably from 2013 to 2018 in the three regions, with a downward trend of -12.2 µg m$^{-3}$ yr$^{-1}$ in BTH, -7.7 µg m$^{-3}$ yr$^{-1}$ in YRD and -5.0 µg m$^{-3}$ yr$^{-1}$ in PRD. Moreover, MLR-EMIS model can mostly capture these decreasing features. The emission-corrected
residual in BTH, YRD and PRD has a decreasing trend of -0.03 µg m$^{-3}$ yr$^{-1}$, -0.1 µg m$^{-3}$ yr$^{-1}$ and 0.2 µg m$^{-3}$ yr$^{-1}$ respectively, or 0.2%, 1.0% and -4.0% of the observed trends, respectively. Hence, the contribution of variations in climate/meteorology to the observed linear trend in winter PM$_{2.5}$ in BTH, YRD and PRD from 2013 to 2018 is 0.2%, 1.0% and -4.0% respectively, meaning that 99.8%, 99.0% and 104.0% of the linear trend in the observed winter PM$_{2.5}$ is attributable to emission (Table 1). These results of MLR-EMIS are in good agreement with Zhai et al. (2019) and Chen et al. (2019). This agreement is
reinforced by a mechanistic model assessment conducted by Chen et al. (2019), who suggested that the contribution of





emissions to the linear trend in PM$_{2.5}$ in 2013–2017 was 79%, while the contribution by meteorology was 21%. Furthermore, Gong et al. (2021) developed a framework based on an Environmental Meteorology Index, to quantitatively assess the contribution of meteorology variations to the trend of PM$_{2.5}$ concentrations and separate the impacts of meteorology from the emission-control measures. They found that emission control contributed more than 90% of the PM$_{2.5}$ decline in BTH from

2013 to 2017, again in good agreement with those values in BTH and Beijing shown in Table 1 (upper two rows).

Wang et al. (2015) hypothesized that decreasing ASI could be an important contributor to the recent increased haze days in eastern China, and about 45–67% of the interannual to interdecadal variability of winter haze days could be explained by the ASI variability. Following the MLR-EMIS model study above, we carry out a parallel MLR-ASI model study in which the emission is replaced by ASI (Figure 3). The ASI-corrected trends differ (i.e., residuals) by -9.3 µg m$^{-3}$ yr$^{-1}$, -3.3 µg m$^{-3}$ yr$^{-1}$

and -3.0 µg m$^{-3}$ yr$^{-1}$, respectively, from the observed linear trends in winter PM$_{2.5}$ in BTH, YRD and PRD, which imply that the emissions are responsible for 76%, 43% and 60% of the observed linear trend in winter PM$_{2.5}$ in BTH, YRD and PRD, respectively, in 2013–2018 (Table 2). It follows that ASI can explain 24%, 57% and 40% of the decreasing trends (Table 2), agreeing with Wang et al. (2015). Here we note that the contribution of ASI represents a partial contribution of climate/meteorological conditions, adding other meteorological parameters would increase the contribution.

The comparison of Tables 1 and 2 poses an interesting problem. For BTH, the MLR-ASI value of 76% (Table 2) for the contribution of emissions is slightly lower than, but nevertheless agrees approximately with the results of MLR-EMIS in the upper two rows of Table 1. However, for YRD and PRD, the contributions of 43% and 60% (Table 2), respectively, by non-ASI (including emissions) to the observed linear trends in winter PM$_{2.5}$ are significantly less than those values of 96%–99% derived by the MLR-EMIS analysis (bottom two rows of Table 1). What causes the discrepancy? Which table has the correct

results? The answer to the first question is that there are significant linear trends in PM$_{2.5}$, anthropogenic emissions and ASI (Figures 1 and 3), but there is no significant linear trend in the meteorological parameters used in the studies by Zhai et al. (2019) and Chen et al. (2019). The MLR analysis gives high values to correlation coefficients of parameters with significant linear trends. In fact, any parameter with a significant linear trend in 2013–2018, e.g., sea surface temperature (SST) of the western Pacific, would get a high correlation coefficient in the MLR analysis.

**3.2 Compare MLR results to mechanistic models**

The answer to the second question is that neither of the two tables is correct for the following reasons: All evaluations in Tables 1 and 2 quantified the relative contributions of anthropogenic emission and meteorology to the linear trend of PM$_{2.5}$ in 2013−2018 using certain statistical MLR models. No mechanistic process was considered in these models. This raises a fundamental concern about the attribution of causes based on statistical regression results alone. It is well known that a good

correlation in the MLR analysis does not imply any causal relationship, let alone a quantitative causal relationship. The causal relationship can only be established if a mechanistic model, that simulates the atmospheric environment with realistic emissions of air pollutants and ambient meteorological conditions as model inputs, can credibly reproduce the observed





concentrations and trends of PM$_{2.5}$. Therefore, we conclude that the quantitative results in Tables 1 and 2 are not scientifically credible unless corroborated by a mechanistic model.

The KZ-MLR results of Chen et al. (2019) in Table 1 are corroborated by the Weather Research and Forecasting and the Community Multi-scale Air Quality model (WRF-CMAQ), a mechanistic model (Chen et al., 2019). However, the mechanistic model study of Chen et al., (2019) is consisted of short term (2013–2017) simulations. As noted in Section 2.2, a short-term study of 2013–2017 overlooks the effect as well as the cause of bulge-2013, which might lead to significant uncertainties and/or biased results. In this context, we believe that an effective mechanistic (or MLR) model study should

cover long enough time before the period of interest (2013–2017), so that the model could be constrained by the major features of interannual variations such as the bulge-2013 and the depression before it (Figure 1a).

We summarize the discussions involving Tables 1 and 2 as follows. (1) Quantitative MLR results in Tables 1 and 2 without corroboration by a mechanistic model are not credible. (2) Results from mechanistic models are more credible than MLR in theory, but significant uncertainties and/or biased results exist in the short term model simulations.

**3.3 An alternative interpretation of MLR results**

In view of the difficulty in interpreting the results from MLR analysis, we propose an alternative interpretation of the correlation coefficient by interpreting it as "the maximum contribution of an independent variable (e.g., emissions in Table 1) to the dependent variable (e.g., linear trends of PM$_{2.5}$ in Table 1)", while the residual should be interpreted as the minimum contribution of all other independent variables (e.g., non-emission variables in Table 1). Tables 3 and 4 are the

corresponding outcomes from the new interpretation for Tables 1 and 2, respectively. It is clear that now under the new interpretation, Tables 3 and 4, unlike Tables 1 and 2, are consistent with each other. In addition, the results of Chen et al. (2019) and Gong et al. (2021) are also consistent with the range of values in Tables 3 and 4. These consistent results provide a firm cornerstone to explore additional applications of the alternative interpretation as discussed in Section 3.4.

Another critical factor affecting the MLR results is the length of time period studied. Table 5 compares the results of two

short-term studies (2013–2017; 2013–2018) in BTH to a long-term study (1985–2018) for MLR-EMIS and MLR-ASI. The emissions of MLR-EMIS can contribute to a maximum of 99% of the observed trend of PM$_{2.5}$ in BTH during 2013–2017, while the residual parameters contribute at least 1%. For the MLR-ASI analysis, the maximum possible contribution of ASI to observed PM$_{2.5}$ in 2013–2017 is 30%, while the residuals contribute at least 70%. Using the retrieved PM$_{2.5}$ does not change the results significantly from the observed PM$_{2.5}$. Adding 2018 into consideration makes little difference for MLR-

EMIS. However, adding only the year 2018, the ASI's maximum contribution to observed PM$_{2.5}$ decline by about 11% compared to that of 2013–2017, while the minimum contribution of the residual increases from 57% to 68% (Table 5). The reason for the larger change in the MLR-ASI can be readily understood by comparing the solid red line (EMIS) in Figure 2a to that (ASI) of Figure 3a, the former maintained a smooth declining trend in 2016–2018 while the latter turned around and increased from 2016 to 2018. This turning-around made the MLR-ASI regression significantly worse and was responsible

for the residual increasing from 57% to 68%.



The 2013–2017 and 2013–2018 results of MLR-EMIS in YRD (Table 6) and PRD (Table 7) are also consistent with those in BTH (Table 5). The 2013–2017 and 2013–2018 results of MLR-ASI in PRD are consistent with those in BTH, while the values of corresponding results of YRD are relatively large compared with the other two regions. The reason for the difference is that the solid red line (ASI) and the solid black line (observed PM$_{2.5}$) in YRD showed relatively consistent trends from 2013 to 2017, while both lines show opposite trends in the 2014–2017 BTH and 2015–2018 PRD (Figure 3). The consistent trend between the two lines makes the MLR-ASI regression in YRD relatively good, resulting in the maximum contribution of ASI to PM$_{2.5}$ in YRD from 2013 to 2017 being 80%, while the residual contribution is greater than 20%. After adding the opposite upward trend of the two curves from 2017 to 2018, the maximum ASI contribution to PM$_{2.5}$ in YRD decreased substantially from 85% to 54%, while the residual increased to >46% (Table 6).

Extending to 1985–2018, these long-term results differ drastically from the short-term results: The contribution of emissions in MLR-EMIS to the linear trends of PM$_{2.5}$ in BTH for the 34-year study is merely <7%, while the contribution of all residual parameters is >93%. The small upper limit of 7% can be easily explained by examining Figure 4a in which the regression between the red line (emission) and black line (PM$_{2.5}$) is very poor, especially after 2010 when the observed PM$_{2.5}$ started to climb from around 20% to the bulge-2013 of 100%. In fact, the red emission line in Figure 1a had to be turned upside down in Figure 4a to get the best fit to the black line of PM$_{2.5}$ (note the fit between 1988 and 2012 is fairly good), which resulted the red line to miss the bulge-2013 completely. The small contribution of emissions compared to the residuals is in good agreement with the results of Dang and Liao (2019) who found that meteorology contributed significantly more than emissions to the linear trend as well as the interannual variability of severe winter haze days in BTH in 2013−2017. The MLR-ASI analysis for PM$_{2.5}$ in BTH over the 34-year period from 1985 to 2018 has a slightly better regression result as shown in Figure 5a. As a result, the maximum contribution of ASI to the linear trend of PM$_{2.5}$ is 43%, while the minimum contribution of the residuals is 57% (Table 5). The bulge-2013 in the winter haze days in North China Plain was also noticed by Yin and Wang (2017), whose generalized additive model using ASI and SST as predictors was found to capture the interannual and interdecadal variations of winter haze days in 1980−2015, including the bulge-2013.

For YRD, the 1985–2018 emission contribution to the linear trend in the observed PM$_{2.5}$ has an upper limit of only 1%, implying the contribution of residuals (including meteorology) to be at least 99% (Table 6). The small 1% can be explained by the extremely poor match between the red line (emission) and the black line (PM$_{2.5}$) in Figure 4b. The regression result of the MLR-ASI analysis in YRD is relatively good. Although the red line (ASI) also fails to match bulge-2013, it matches well with the 2016−2018 profile as shown in Figure 5b. As a result, the maximum contribution of ASI to the 34-year linear trend of PM$_{2.5}$ reaches 75%, implying the minimum contribution of the residuals to be 25% (Table 6). PRD does not have the bulge-2013 (Figure 1c) and therefore has a good regression between emissions and PM$_{2.5}$ from 1985 to 2018 (Figure 4c), implying that emissions over 34-year period can contribute as much as 73% to the linear trend of PM$_{2.5}$, while >27% is contributed by the residuals (Table 7). The regression result of the MLR-ASI analysis for PM$_{2.5}$ in PRD from 1985 to 2018 is slightly better than those of the MLR-EMIS, and the red line (ASI) and the black line (PM$_{2.5}$) show opposite trends only in





2011−2015 as shown in Figure 5c, resulting in the maximum contribution of ASI to the linear trend of PM$_{2.5}$ being 81%,
while the minimum contribution of the residuals is 19% (Table 7).

### 3.4 Best estimate of the contribution

Tables 5−7 and Figures 4−5 provide strong evidence supporting the notion that the contributions of emissions and meteorology to the linear trend in PM$_{2.5}$ depend on the length of time analyzed. A critical question remaining is which regression analysis gives the correct value of the contribution when a long-term analysis overlaps with a short-term analysis,
e.g., the 1985−2018$^{ret}$ analysis (Figure 4a) vs. the 2013−2018$^{ret}$ analysis (Figure 2a) for BTH during the period 2013−2018 (fourth row vs. last row in Table 5)? A logical answer to this question is that the long-term analysis gives the correct value because it has more data points to constrain the regression. This is discussed in the following using the BTH case as an example.

Figure 4a-a shows an enlarged plot of the 2013−2018 portion of Figure 4a (named as 2013−2018$^{ret34}$ analysis hereafter and in
Tables 5−7). The green solid bars in Figure 4a-a denote the anomalies/deviations of the red emission line from the black observations line, of which the mean absolute value of 49% (third column and the last row of Table 5) can be recognized as the minimum contribution of the residual/non-emission (including meteorology/climate) to the linear trend in PM$_{2.5}$ according to the alternative interpretation. Hence the maximum contribution by EMIS is 51%, which is substantially less than the 94% of the 2013−2018$^{ret}$ analysis (second column and fourth row of Table 5). The main reason for this difference
can be traced to the bulge-2013 in Figure 4a which, as discussed in Section 3.3, contributes pivotally to the red emission line in Figure 1a being turned upside down in Figure 4a to get the best fit with the black line of observed PM$_{2.5}$. For the 2013−2018$^{ret34}$ ASI analysis, the green solid bars of Figure 5a-a denote the anomalies/deviations of ASI line from the PM$_{2.5}$ line, of which the mean absolute value of 38% (fifth column and last row of Table 5) can be recognized as the minimum contribution of the residual of ASI, which includes emissions. Hence, we derive the maximum contribution by ASI to be 62%
(fourth column and last row of Table 5). Key results of the two analyses above are summarized in the fourth and last rows of Table 5. Qualitatively the results of the 2013−2018$^{ret}$ analysis and the 2013−2018$^{ret34}$ analysis are consistent (overlap) with each other, quantitatively a significant difference exists between the two analyses-the contribution of emission to the linear trend in PM$_{2.5}$ in the latter has a tighter upper limit of only 51% compared to 94% of the former, and a greater lower limit for the contribution of the residual which includes meteorology/climate in the latter (49%) compared to the former (6%). The
fact that the long-term analysis renders a tighter upper/lower limit is evidently the result of additional constraints provided by the long-term data. Finally, since a tighter upper/lower limit gives a more precise estimate of the contribution, we suggest that the best estimates of the contributions of emission, ASI and other meteorology/climate parameters to the linear trend in PM$_{2.5}$ in BTH during 2013−2018 are those listed in the last row of Table 5, specifically, emission<51%, non-emission>49%, ASI<62% and non-ASI>38%. These are our best estimates which are remarkably different from those of MLR studies listed
in Table 1.



The same analysis can be carried out for YRD and PRD, and the key results are summarized in the fourth and last rows of Tables 6 and 7, respectively. For YRD, the best estimates of the contributions of emission, ASI and other meteorology/climate parameters to the linear trend in PM$_{2.5}$ during 2013−2018 are those listed in the last row of Table 6, specifically, emission<44%, non-emission>56%, ASI<74% and non-ASI>26%. For PRD, the best estimates of the
contributions of emission, ASI and other meteorology/climate parameters to the linear trend in PM$_{2.5}$ during 2013−2018 are those listed in the last row of Table 7, specifically, emission<88%, non-emission>12%, ASI<55% and non-ASI>45%. These best estimates are also significantly different from those of MLR studies listed in Table 1.

## 4 Summary and conclusions

Recently, Chen et al. (2019) and Zhai et al. (2019) used MLR models to analyze the significant downward trend of PM$_{2.5}$
concentrations in China's major air pollution regions in 2013–2018 (2013–2017 for Chen et al. (2019)) and quantified that the control of anthropogenic emissions accounted for 81% to 103% of the PM$_{2.5}$ reduction (Table 1). While there is little doubt that anthropogenic emissions make a significant contribution to the reduction trend of PM$_{2.5}$, we are skeptical of these high contributions by emissions obtained based solely on MLR models. In fact, a good correlation in MLR analysis does not imply any causal relationship, let alone a quantitative relationship. The causal relationship can only be established if a
mechanistic model, that simulates the atmospheric environment with realistic emissions of air pollutants and ambient meteorological conditions as model inputs, can credibly reproduce the observed concentrations and trends of PM$_{2.5}$. In this regard, Chen et al. (2019) corroborated their MLR result of 81% (Table 1) using the mechanistic model WRF-CMAQ. However, the mechanistic model study of Chen et al., (2019) is consisted of short term (2013–2017) simulations. As noted in Section 2.2, a short-term study of 2013–2017 overlooks the effect as well as the cause of bulge-2013, which might lead to
significant uncertainties and/or biased results. In this context, we believe that an effective mechanistic (or MLR) model study should cover long enough time before the period of interest (2013–2017), so that the model could be constrained by the major features of interannual variations such as the bulge-2013 and the depression before it (Figure 1a).

To compare with previous MLR studies, the MLR model is used in this study to assess the contributions of climate/meteorology variations and anthropogenic emissions to the linear trends of PM$_{2.5}$ concentration in three regions in
eastern China, namely BTH, YRD, and PRD. We first carry out an MLR analysis (MLR-EMIS) on the emissions and observed trend of PM$_{2.5}$ in BTH during 2013−2018, and show that the results of Zhai et al. (2019) and Chen et al. (2019) can be satisfactorily reproduced (Table 1). Then the same MLR analysis are performed on the arctic sea ice (ASI) and observed trend of PM$_{2.5}$ (MLR-ASI), and obtain a 76% contribution by emissions in BTH, which is slightly lower than, but nonetheless in qualitative agreement with those values obtained by Zhai et al. (2019) and Chen et al. (2019) (Tables 1 and 2).
However, for YRD and PRD, the contributions of emissions to the observed trends of PM$_{2.5}$ are only 43% and 60% (Table 2), respectively, significantly less than those values of 96%−99% derived by the MLR-EMIS analysis (Table 1). We believe that the discrepancy is rooted in the false assumption/interpretation of the correlation coefficient as the value of contribution in



the MRL studies. This assumption/interpretation is conceptually false because a good correlation in a MLR analysis does not imply a causal relationship, and certainly not a quantitative relationship. We therefore propose an alternative interpretation:

the correlation coefficient should be interpreted as the maximum possible contribution of the independent variable to the dependent variable, and the residual should be interpreted as the minimum contribution of all other independent variables. Under the new interpretation, the new results, as shown in Tables 3–4, become consistent with one another.

Another important outcome from this study is that the results of a short-term (2013–2018) analysis are significantly different from those of a long-term (1985–2018) analysis for the period 2013−2018 they overlap, suggesting that MLR results depend

critically on the length of time analyzed. The long-term analysis renders a more precise estimate, because of additional constraints provided by the long-term data. We therefore suggest that the best estimates of the contributions of emissions and non-emission (including climate/meteorology) to the linear trend in $PM_{2.5}$ during 2013–2018 are those from the long-term analyses: i.e., emission <51% and non-emission >49% for BTH, emission <44% and non-emission >56% for YRD, emission <88% and non-emission >12% for PRD.


*Data availability.* Visibility data were obtained from Global Summary of Day (GSOD) provided by the National Climatic Data Center (NCDC) (https://gis.ncdc.noaa.gov/maps/ncei/cdo/daily, last access: 10 March 2022). Surface $PM_{2.5}$ measurements from 2013–2019 are taken from China National Environment Monitoring Center (CNEMC, http://www.cnemc.cn/, last access: 10 March 2022). The data of this paper are available upon request to Shaw Chen Liu

(shawliu@jnu.edu.cn).

*Author Contributions.* RL and SL proposed the essential research idea. YW performed the analysis. YW, RL and SL drafted the manuscript. JD, YL, ZH and JZ helped analysis and offered valuable comments. All authors have read and agreed to the published version of the manuscript.


*Competing interests.* The authors declare that they have no conflict of interest.

*Acknowledgments.* The authors thank National Climatic Data Center (NCDC) and the China National Environmental Centre for providing datasets that made this work possible. We also acknowledge the support of the Institute for Environmental and

Climate Research and Guangdong-Hong Kong-Macau Joint Laboratory of Collaborative Innovation for Environmental Quality in Jinan University.

*Financial support.* This work was jointly supported by the National Natural Science Foundation of China (92044302, 41805115), Guangzhou Municipal Science and Technology Project, China (202002020065), Special Fund Project for

Science and Technology Innovation Strategy of Guangdong Province (2019B121205004), and Guangdong Innovative and Entrepreneurial Research Team Program (2016ZT06N263).



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



**Table 1: Comparison of the contribution of emissions and meteorological conditions to the observed PM$_{2.5}$ trends in 2013−2018 (2013−2017 for Chen et al. (2019)).**

| Research | Region | Method | Emission | Meteorology |
|---|---|---|---|---|
| Chen et al. (2019) | Beijing | KZ-MLR | 81% | 19% |
| Zhai et al. (2019) | BTH | MLR | 86% | 14% |
| This study | | MLR-EMIS | 99.8% | 0.2% |
| Zhai et al. (2019) | YRD | MLR | 103% | -3% |
| This study | | MLR-EMIS | 99% | 1% |
| Zhai et al. (2019) | PRD | MLR | 81% | 19% |
| This study | | MLR-EMIS | 96% | 4% |

Note: KZ: Kolmogorov–Zurbenko filter; MLR: stepwise multiple linear regression; MLR-EMIS indicates the use of emissions and PM$_{2.5}$ as inputs to the MLR model.




**Table 2: Contributions of arctic sea ice (ASI) and residuals (Non-ASI) to the observed PM$_{2.5}$ trends in the winter seasons of 2013−2018.**

| Region | Method | ASI | Non-ASI |
|--------|--------|-----|---------|
| BTH | | 24% | 76% |
| YRD | MLR-ASI | 57% | 43% |
| PRD | | 40% | 60% |

Note: MLR-ASI indicates the use of ASI and PM$_{2.5}$ as inputs to the MLR model.



**Table 3. Contributions of emissions and non-emissions (including climate and meteorological conditions) to the observed PM$_{2.5}$ trends in 2013−2018 (2013−2017 for Chen et al. (2019)).**

| Research | Region | Method | Emission | Non-emission |
|---|---|---|---|---|
| Chen et al. (2019) | Beijing | KZ-MLR | <81% | >19% |
| Zhai et al. (2019) | BTH | MLR | <86% | >14% |
| This study | | MLR-EMIS | <99.8% | >0.2% |
| Zhai et al. (2019) | YRD | MLR | <103% | >-3% |
| This study | | MLR-EMIS | <99% | >1% |
| Zhai et al. (2019) | PRD | MLR | <81% | >19% |
| This study | | MLR-EMIS | <96% | >4% |





**Table 4. Contributions of ASI and residuals (Non-ASI) to the observed PM$_{2.5}$ trends in 2013−2018.**

| Region | Method | ASI | Non-ASI |
|--------|--------|-----|---------|
| BTH | | <24% | >76% |
| YRD | MLR-ASI | <57% | >43% |
| PRD | | <40% | >60% |





**Table 5. Contributions of emissions and ASI to the PM$_{2.5}$ trends in BTH.**

| Time Period | MLR-EMIS | | MLR-ASI | |
|---|---|---|---|---|
| | Emission | Non-emission | ASI | Non-ASI |
| 2013−2017[obs] | <99% | >1% | <30% | >70% |
| 2013−2017[ret] | <94% | >6% | <43% | >57% |
| 2013−2018[obs] | <99.8% | >0.2% | <24% | >76% |
| 2013−2018[ret] | <94% | >6% | <32% | >68% |
| 1985−2018[ret] | <7% | >93% | <43% | >57% |
| 2013−2018[ret34] | <51% | >49% | <62% | >38% |

Note: The superscripts obs and ret indicate the use of observed and retrieved PM$_{2.5}$ data, respectively. Superscript ret34 indicates the retrieved PM$_{2.5}$ data for 2013–2018 obtained using the data of 1985–2018.





**Table 6. Contributions of emissions and ASI to the PM$_{2.5}$ trends in YRD.**

| Time Period | MLR-EMIS | | MLR-ASI | |
|---|---|---|---|---|
| | Emission | Non-emission | ASI | Non-ASI |
| 2013−2017[obs] | <96% | >4% | <80% | >20% |
| 2013−2017[ret] | <84% | >16% | <85% | >15% |
| 2013−2018[obs] | <99% | >1% | <57% | >43% |
| 2013−2018[ret] | <86% | >14% | <54% | >46% |
| 1985−2018[ret] | <1% | >99% | <75% | >25% |
| 2013−2018[ret34] | <44% | >56% | <74% | >26% |






**Table 7. Contributions of emissions and ASI to the PM$_{2.5}$ trends in PRD.**

| Time Period | MLR-EMIS | | MLR-ASI | |
|---|---|---|---|---|
| | Emission | Non-emission | ASI | Non-ASI |
| 2013−2017[obs] | <81% | >19% | <67% | >33% |
| 2013−2017[ret] | <96% | >4% | <59% | >41% |
| 2013−2018[obs] | <99.96% | >0.04% | <40% | >60% |
| 2013−2018[ret] | <99.2% | >0.8% | <39% | >61% |
| 1985−2018[ret] | <73% | >27% | <81% | >19% |
| 2013−2018 [ret34] | <88% | >12% | <55% | >45% |

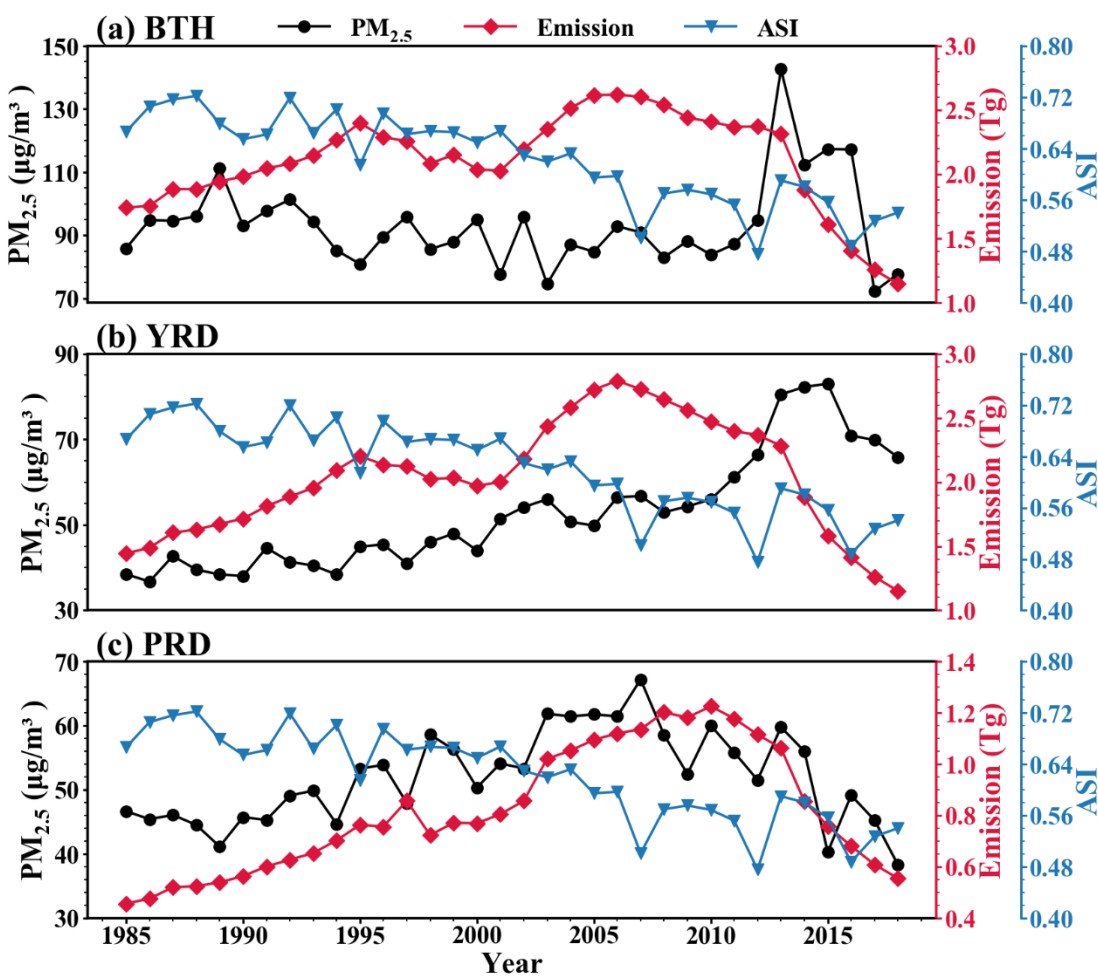

**Figure 1: PM$_{2.5}$ concentration, emission and Arctic Sea Ice (ASI) in BTH, YRD and PRD in the winter seasons of 1985–2018.**





**Figure 2: Results of MLR-EMIS analysis for 2013–2018 in BTH (a), YRD (b) and PRD (c). Temporal variations of observed winter PM₂.₅ concentration are shown in black, contributions of anthropogenic emissions to the PM₂.₅ trend are shown in red, and the residual is shown in blue. Values inset in each panel are the ordinary linear regression trends, with 95% confidence intervals obtained by the student's t test.**






**Figure 3: Same as Figure 2 except for MLR-ASI analysis.**

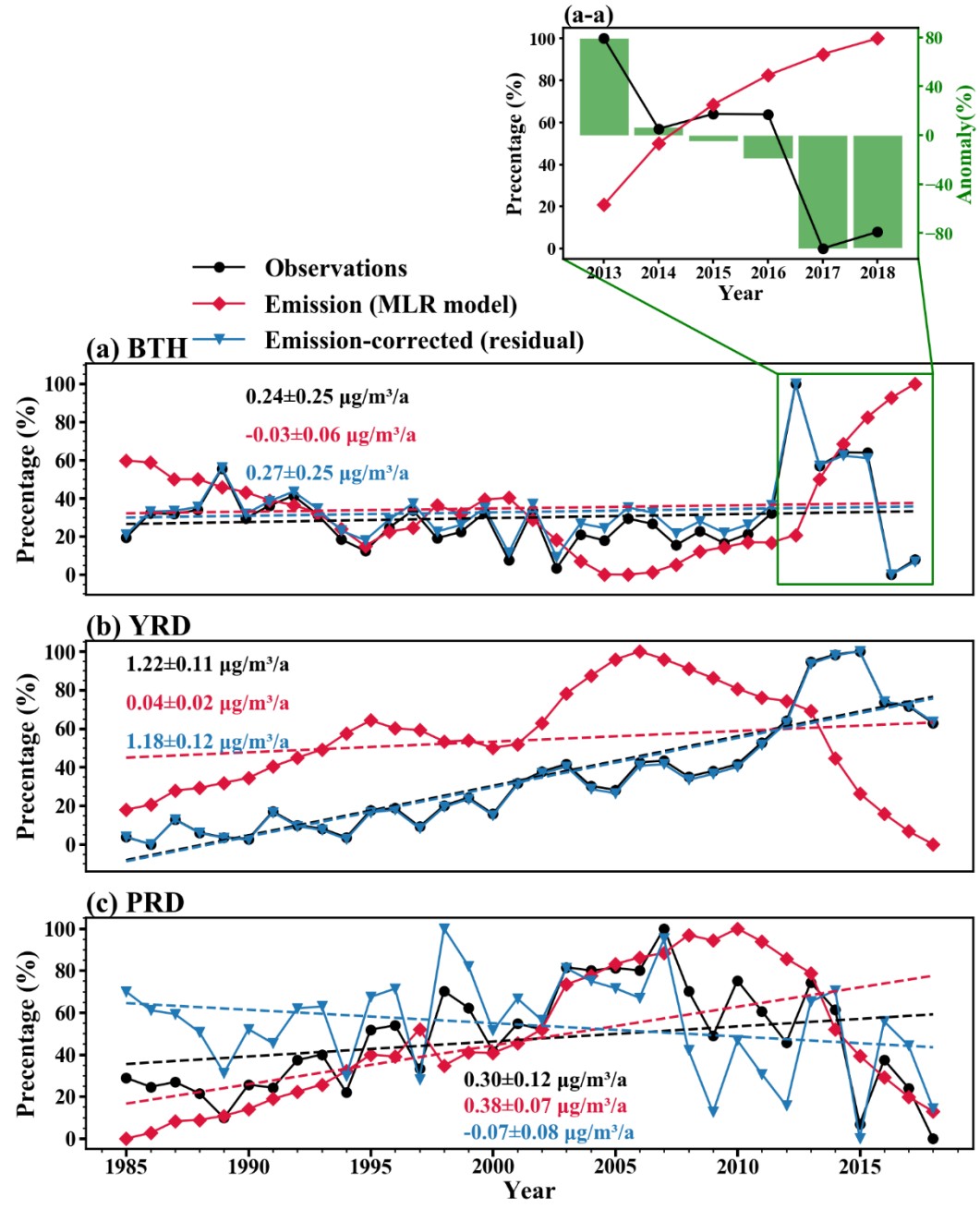

**Figure 4: The same as Figure 2 except for time period of 1985–2018. Subfigures 4(a-a) are enlarged schematic representations of the period 2013–2018 in Figure 4(a).**


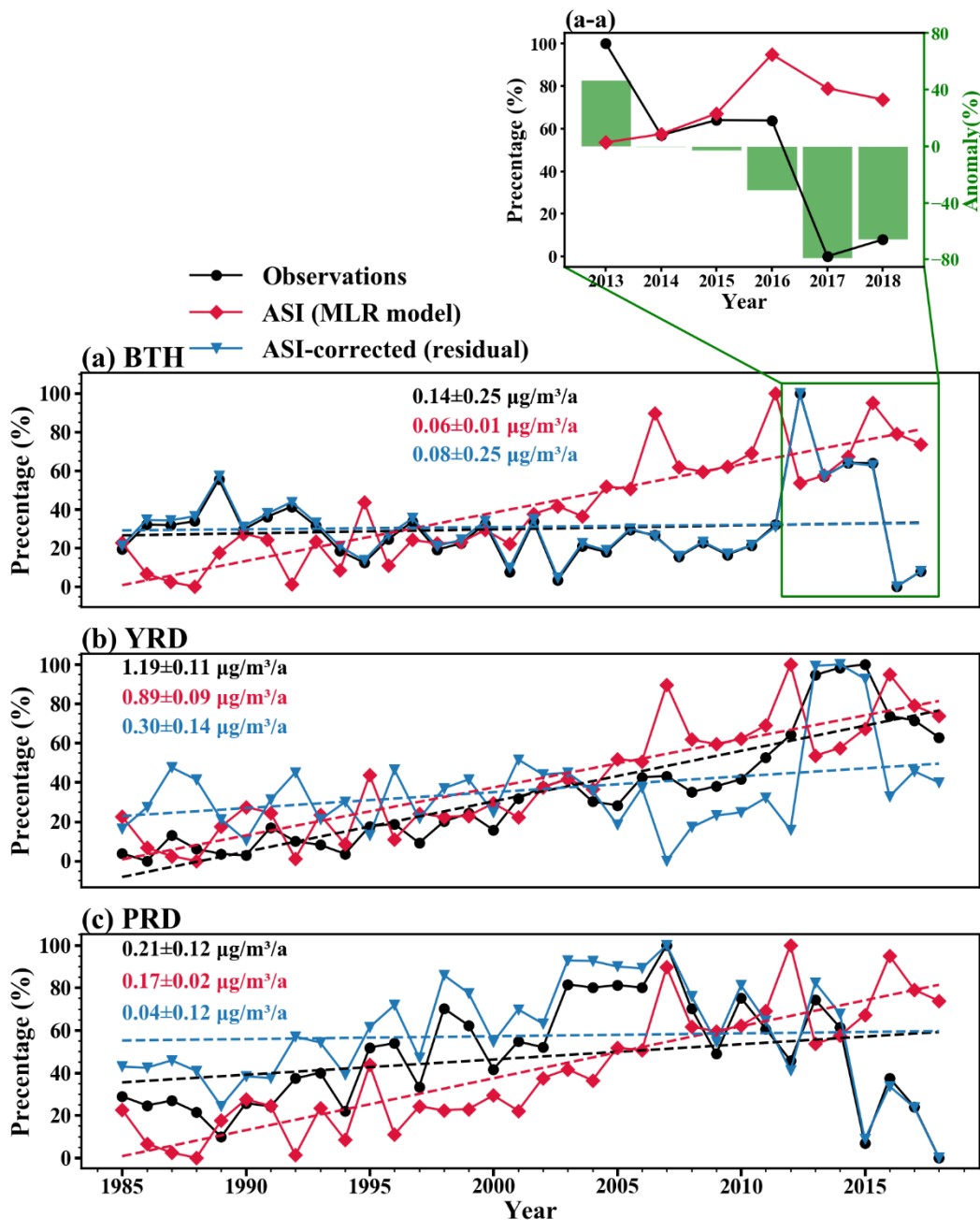

**Figure 5: The same as Figure 4 except for MLR-ASI analysis.**