# Peer review of "Contributions of meteorology and anthropogenic emissions to the trends in winter PM2.5 in eastern China 2013–2018"

_Atmospheric Chemistry and Physics, 2022_

## Community Comment (CC1)

Dear Editor,

We appreciate the prompt reviews and would like to thank the reviewer for insightful comments and suggestions on our manuscript entitled "Contributions of meteorology and anthropogenic emissions to the trends in winter $PM_{2.5}$ in eastern China 2013–2018" (MS No.: acp-2022-304). We have carefully considered all comments and suggestions. Listed below are our point-by-point responses to all comments and suggestions of this reviewer (Reviewer's points in black, our responses in blue).

**Anonymous Referee #1**

This paper presents a MLR statistical attribution of the 1985-2018 $PM_{2.5}$ trends in three megacity clusters in China, using visibility data as proxy for pre-2013 $PM_{2.5}$ data. It finds a large meteorological (non-emission) contribution to the trend, and argues that previous MLR analyses of the 2013-2018 trend using the actual $PM_{2.5}$ data starting in 2013 and attributing the trend to emissions are not robust. The paper makes some good points about the difficulty of sorting out meteorological effects when interpreting short (post-2013) trends. However, I believe that it may be (1) flawed in its reconstruction of the 1985-2018 $PM_{2.5}$ record which is the basis for most of the argumentation, (2) mistaken in claiming that attribution of recent $PM_{2.5}$ trends to emissions is not based on mechanistic knowledge, and (3) annoying in belaboring trivial statistical points that are well known to any trained scientist. I don't think that this paper is publishable in ACP in current form.

**Response:**

The reviewer made three general criticisms. Our responses are listed below one by one.

(1) Our reconstruction of the 1985–2018 $PM_{2.5}$ record is based on a method that converts observed visibility data to the concentrations of $PM_{2.5}$. This method has been shown in many previous studies to be credible (Shen et al., 2016; Liu et al., 2017; Gui et al., 2020; Li et al., 2020; Li et al., 2021). The first two references are already cited in

our paper. Given the serious concern of this reviewer, in Figure R1 below we compare the winter $PM_{2.5}$ record derived in our study (black line) to winter haze days derived from observed visibility data in Beijing by Li et al. (2021) (green line). The winter $PM_{2.5}$ concentrations are expected to be well correlated with the number of winter haze days. Indeed the correlation coefficient between the green line and black line is quite high at 0.7. Also shown in Figure 1 are $PM_{2.5}$ concentrations observed by the US Embassy in Beijing (blue line, 2009–2018) and those observed by CNEMC in BTH (red line, 2013–2018). The crucial bulge-2013 shows up consistently in all data sets. The correlation coefficients between our $PM_{2.5}$ values and those of the US Embassy and CNEMC are also very high at 0.6 and 0.9, respectively. These high correlation coefficients suggest that our reconstruction of the 1985–2018 $PM_{2.5}$ record from observed visibility data is credible.

[Figure]

Figure R1. Temporal variations of winter inversed $PM_{2.5}$ concentrations in BTH of this study (black, 1985–2018), simulated $PM_{2.5}$ concentrations in Beijing by Dang and Liao (2019) (purple, 1985–2017), $PM_{2.5}$ concentrations observed by the US Embassy in Beijing (blue, 2009–2018) and those observed by CNEMC in BTH (red, 2013–2018).

(2) In our paper we didn't state "that attribution of recent $PM_{2.5}$ trends to emissions is not based on mechanistic knowledge". What we did state was that *quantitative attribution* of recent $PM_{2.5}$ trends to emissions is not based on *realistic/credible mechanistic models*. There is a significant difference between mechanistic knowledge

and realistic/credible mechanistic models. Only realistic/credible mechanistic models have the capability of making quantitative attribution of recent $PM_{2.5}$ trends. However, it is extremely formidable to make a multi-year realistic/credible simulation of the winter mean $PM_{2.5}$ in the megacity clusters in China. In our opinion, the most realistic multi-year mechanistic model simulation is the study by Dang and Liao (2019), who made a 33-year (1985–2017) model simulation study of severe winter haze days in BTH (purple line in Figure R1). There is an excellent agreement between the purple line and $PM_{2.5}$ concentrations observed by the US Embassy in Beijing (blue line, 2009–2018). The agreement with $PM_{2.5}$ concentrations observed by CNEMC in BTH (red line, 2013-2018) is also very good. For the entire period of 1985–2017, there are moderate mismatches near 1997–2002 and 2010 between the purple line (Dang and Liao, 2019) and green line (Li et al., 2021), but still has an acceptable overall correlation coefficient of 0.4. As cited in lines 201–202 of our paper, Dang and Liao (2019) "found that meteorology contributed significantly more than emissions to the linear trend", which is consistent with the result of our study.

(3) We accept the criticism of "belaboring trivial statistical points that are well known to any trained scientist." We will delete some of the repeated statements in the revised manuscript. We were surprised that previous MLR studies would have overlooked these trivial yet important statistical points, and tried to find an explanation. That leads us to realize the importance of the bulge-2013 (as noted by this reviewer in specific comment #1 below) and to suggest the "maximum possible contribution" as an alternative interpretation for the MLR results.

**Specific comments**

1. The 'bulge-2013' feature in Figure 1 (line 88) anchors much of the argumentation in the paper but it is very weird. It seems caused by the switch from the visibility proxy to the actual $PM_{2.5}$ data in 2013. The methods are buried in Supplementary Material. Is this 'bulge-2013' seen in the consistent long-term satellite AOD data record? I think that the authors would have to show that it is present in the AOD data in order to have

credibility.

**Response:**

We believe that Figure R1 and associated discussions above address this comment adequately. In response to the question about AOD, we compare below the winter satellite AOD data (MERRA2) in BTH to $PM_{2.5}$ and visibility (both from this study) in Figure R2. The correlation between AOD and $PM_{2.5}$ is fair (overall correlation coefficient 0.3) except some mismatches during two periods (2007–2009 and 2012–2013). As a result, only half of the bulge (2013–2018) can be seen in AOD. The reason for the mismatches is probably because surface $PM_{2.5}$ is sensitive to the height of mixed layer while AOD is not. In other words, changes of surface $PM_{2.5}$ due to changing mixing height are usually not detected in the AOD observations.

[Figure]

Figure R2. Time series of winter $PM_{2.5}$ concentration, visibility (both from our study) and AOD (from MERRA2) in BTH from 1985 to 2018.

2. What is the 'emission' in Figure 1? Of what species?

**Response:**

The 'emission' is composed of $PM_{10}$, $PM_{2.5}$, $SO_2$, $NH_3$, NOx, black carbon, and organic carbon in three sets of emission inventories (PKU inventory, MEIC inventory and PRD-EI inventory). Data and calculation methods for emissions are presented in Section 2.1.

As an example, Figure R3 shows the temporal variation of three emission inventories in PRD. They show generally consistent variations during overlapping periods.

[Figure]

Figure R3. PKU emission inventory for winter 1985–2012, MEIC emission inventory for winter 2010–2016 and PRD-EI emission inventory for winter 2006–2018 for PRD. The raw data are normalized to the difference of the maximum value and minimum value.

3. Line 91: the Mao et al. 2019 reference which is intended to provide support for the authors' meteorological attribution of the trend is in fact grey literature involving some of the same authors.

**Response:**

Mao et al. (2019) is a peer reviewed article, not a "grey literature". The Mao et al. (2019) reference (Lines 352–353) is reproduced below:

Mao, L., Liu, R., Liao, W., Wang, X., Shao, M., Liu, S. C. and Zhang, Y.: An observation-based perspective of winter haze days in four major polluted regions of China, Natl. Sci. Rev., 6(3), 515–523, https://doi.org/10.1093/nsr/nwy118, 2019.

National Science Review is published by Oxford University Press on behalf of China Science Publishing & Media Ltd. The current impact factor of National Science Review is over 23, which ranks it among the best international scientific journals.

4. Line 132, etc.: the mechanistic meteorological connection of ASI to $PM_{2.5}$ is not clear, and as the authors point out any meteorological variable with a suitable long-term trend

would do the trick. But there is in fact a strong mechanistic argument for emissions to be related to $PM_{2.5}$ (line 154), and there is strong independent evidence that Chinese emissions have decreased over the 2013-2018 period (emission inventories, satellite data). To claim that the connection of $PM_{2.5}$ to emissions has no mechanistic support strikes me as obviously wrong. In fact the authors cite Chen et al. 2019 in demonstrating the mechanistic connection in WRF-CMAQ but argue that the analysis is flawed because it did not consider the effect of the bulge-2013 (line 158). As pointed out above, I am very suspicious of this bulge-2013.

**Response:**

We have addressed extensively the issues raised here in general comment #2 (and #1 about the bulge-2013). Moreover, in lines 261–262 we already stated "there is little doubt that anthropogenic emissions make a significant contribution to the reduction trend of $PM_{2.5}$." We were only "skeptical of *those high* contributions by emissions obtained based solely on MLR models."

5. There is a lot of trivial stuff about the non-mechanistic basis of statistical models, correlation not implying causality, more years increasing the credibility of the model, etc., that is repeated again and again and does not rise above the level of a basic course in statistics.

**Response:**

As stated in our response to general comment #3, we accept the criticism of "belaboring trivial statistical points that are well known to any trained scientist," We will delete some of the repeated statements in the revised manuscript.

**References**

Dang, R. and Liao, H.: Severe winter haze days in the Beijing-Tianjin-Hebei region from 1985 to 2017 and the roles of anthropogenic emissions and meteorology, Atmos. Chem. Phys., 19(16), 10801–10816, https://doi.org/10.5194/acp-19-10801-2019, 2019.

Gui, K., Che, H., Zeng, Z., Wang, Y., Zhai, S., Wang, Z., Luo, M., Zhang, L., Liao, T., Zhao, H., Li, L., Zheng, Y. and Zhang, X.: Construction of a virtual $PM_{2.5}$ observation network in China based on high-density surface meteorological observations using the Extreme Gradient Boosting model, Environ. Int., 141, 105801, https://doi.org/10.1016/j.envint.2020.105801, 2020.

Li, H., Yang, Y., Wang, H., Li, B., Wang, P., Li, J. and Liao, H.: Constructing a spatiotemporally coherent long-term $PM_{2.5}$ concentration dataset over China during 1980–2019 using a machine learning approach, Sci. Total Environ., 765, 144263, https://doi.org/10.1016/j.scitotenv.2020.144263, 2021.

Li, S., Chen, L., Huang, G., Lin, J., Yan, Y., Ni, R., Huo, Y., Wang, J., Liu, M., Weng, H., Wang, Y. and Wang, Z.: Retrieval of surface $PM_{2.5}$ mass concentrations over North China using visibility measurements and GEOS-Chem simulations, Atmos. Environ., 222, 117–121, https://doi.org/10.1016/j.atmosenv.2019.117121, 2020.

Liu, M., Bi, J. and Ma, Z.: Visibility-based $PM_{2.5}$ concentrations in China: 1957–1964 and 1973–2014, Environ. Sci. Technol., 51(22), 13161–13169, https://doi.org/10.1021/acs.est.7b03468, 2017.

Mao, L., Liu, R., Liao, W., Wang, X., Shao, M., Liu, S. C. and Zhang, Y.: An observation-based perspective of winter haze days in four major polluted regions of China, Natl. Sci. Rev., 6(3), 515–523, https://doi.org/10.1093/nsr/nwy118, 2019.

Shen, Z., Cao, J., Zhang, L., Zhang, Q., Huang, R. J., Liu, S., Zhao, Z., Zhu, C., Lei, Y., Xu, H. and Zheng, C.: Retrieving historical ambient $PM_{2.5}$ concentrations using existing visibility measurements in Xi'an, Northwest China, Atmos. Environ., 126, 15–20, https://doi.org/10.1016/j.atmosenv.2015.11.040, 2016.

---

## Community Comment (CC2)

Dear Editor,

We appreciate the prompt reviews and would like to thank the reviewer for insightful comments and suggestions on our manuscript entitled "Contributions of meteorology and anthropogenic emissions to the trends in winter $PM_{2.5}$ in eastern China 2013–2018" (MS No.: acp-2022-304). We have carefully considered all comments and suggestions. Listed below are our point-by-point responses to all comments and suggestions of this reviewer (Reviewer's points in black, our responses in blue).

**Anonymous Referee #3**

This work proposes a different method for the MLR analysis of $PM_{2.5}$. Based on the new interpretation and the comparison with previous studies, the MLR results among different studies were found to be more consistent. In addition, the authors also pointed out that the relationship constrained by long-term data is more reliable. Overall, this is an interesting study and it provides some useful information for other researchers when choosing MLR for air quality trend analysis. However, more explanations, especially for the methodology, are still needed.

**Response:**

We appreciate the insightful comments and suggestions. More explanations have been added in our revised manuscript, especially in the methodology section.

**Specific comments**

(1) Line 60, the resolution of the PRD emission inventory is three degrees, which is rather coarse.

**Response:**

The emission inventory of PRD (PRD-EI) is adopted from Huang et al. (2021) and Zhong et al. (2018). Although the resolution of the PRD-EI is coarser than other two inventories, we can only get the emission information of year 2018 and 2019 from PRD-

EI. In addition, in our study, we mainly focus on the long-term trend and interannual variation in the annual total emissions in each region, which is independent of the resolution of emission inventories.

Figure R1 shows the temporal variation of three emission inventories. They have similar variation for the overlapping period.

[Figure]

Figure R1. PKU emissions inventory for winter 1985–2012, MEIC emissions inventory for winter 2010–2016 and PRD-EI emissions inventory for winter 2006–2018 for PRD. The raw data is normalized by the difference of the maximum value and minimum value.

(2) This work mainly focuses on the PRD, YRD and Jing-Jin-Ji regions. For YRD and Jing-Jin-Ji regions, the authors combined the MEIC and PKU emission inventories to do the analysis. While for PRD, they combined PKU and PRD-EI to do the scaling. MEIC and PRD-EI are different emission inventories and the methods that used to derive these two emission inventories should be not consistent. Based on the literatures, the MEIC emission inventory should have already covered the PRD region, why not also using the MEIC emission inventory to analyze the PRD region?

**Response:**

The MEIC inventory does also cover the PRD region, but the time span of MEIC inventory is 2010–2017. The time span of the PRD-EI inventory and PKU inventory is

2006–2019 and 1960–2014, repetitively. Therefore, we combined PRD-EI and PKU inventories in PRD for the winters of 1985–2018.

(3) Please label the scaling factor and Ei equations.

**Response:**

Thanks, we have labeled the scaling factor and $E_i$ equations in our revised manuscript.

(4) Line 71, please use data or reference to support this assumption.

**Response:**

Thanks, we have added Figure S5 in the revised Supplementary Material and revised the Line 71 statement to make it clearer as "Since the ratios of annual emission inventory in PRD to those of YRD and BTH are not expected to change significantly in one or two years (Figure S5)"

[Figure]

Figure S5. Time series of emission inventory (EI) ratios in the winter of 1985–2018 for the BTH/PRD and YRD/PRD, respectively.

(5) The PRD scaling factor was calculated by the emission sum from 2006 to 2013, while the scaling factors for the other two regions were calculated by the emission sum from 2010 to 2013. Please explain why using different emission sum to derive the scaling factors.

**Response:**

We calculate the scaling factor based on the overlapping periods of two inventories. As stated in our response to your Point #2, the time spans of these three inventories are different, so we derived scaling factors for PRD from 2006 to 2013, while for BTH and YRD from 2010 to 2013.

(6) The authors applied the nonlinear exponential fitting to retrieve the long-term $PM_{2.5}$ concentration before 2013, because China began to release the air quality observation data since 2013 and it is unlikely to acquire long-term observation data in this nation before 2013. However, based on the figures in the supplemental material, some of the fittings are not acceptable for further analysis, such as BTH-RH (40, 60) and YRD-RH (90, 100). The authors need to analyze and discuss whether such errors can influence their conclusion.

**Response:**

We believe that our $PM_{2.5}$ retrievals from nonlinear exponential fitting are acceptable for further analysis and do not influence our conclusion for the following reasons: (1) Yes, the correlation coefficient (R) of 0.56 for BTH-RH (40, 60) is a little low, but the fittings for other intervals with more samples are very good, so the overall fitting for BTH remains good. The sample size of YRD-RH (90, 100) is the smallest among all intervals, so its relatively small R (0.36) has negligible effect on the overall fitting for YRD. (2) We compared the retrieved $PM_{2.5}$ concentrations with the observed $PM_{2.5}$ concentration in BTH and YRD (Figure R2), and found that R is more than 0.87, and normalized mean bias (NMB) are 5.9% and 4.6% in BTH and YRD, respectively. These values of R and NMB suggest that the exponential fitting model is capable of reproducing the observed $PM_{2.5}$ concentrations. (3) As you suggested in comment point #7, we use the data of 2015–2019 for the fitting and the 2014 data for the verification, the R (NMB) between the fitted $PM_{2.5}$ concentrations and observed $PM_{2.5}$ concentrations is 0.77 (14.8%), 0.84 (5.5%), and 0.93 (5.1%) in BTH, YRD and PRD,

respectively, suggesting that the exponential fitting models are robust. (4) R between the long-term retrieved PM$_{2.5}$ concentrations and observed visibility reached -0.9 in both BTH and YRD, reconfirming that the performance of our exponential fitting model is satisfactory (Figures S4a-b).

[Figure]

Figure R2. Temporal variation of retrieved PM$_{2.5}$ and observed PM$_{2.5}$ from 2015 to 2018.

(7) For the PM$_{2.5}$ concentration retrieval, I suggest the authors use the data of 2014-2018 for the fitting and the 2013 data for the verification, this can help to verify whether the methods implemented by the authors are reliable or not.

**Response:**

Since we did not have daily PM$_{2.5}$ concentration data for the three regions in 2013, we used observed daily visibility and PM$_{2.5}$ concentration from 2015 to 2019 to establish the exponential fitting model, as shown in Figures S1–S3. Furthermore, we use 2014 daily fitted PM$_{2.5}$ and observed PM$_{2.5}$ for verification (Figure R3). The R (NMB) between the fitted PM$_{2.5}$ concentration and observed PM$_{2.5}$ concentration is 0.77

(14.8%), 0.84 (5.5%), and 0.93 (5.1%) in BTH, YRD and PRD, respectively, verifying that the exponential fitting models are reliable.

[Figure]

Figure R3. Temporal variation of retrieved PM$_{2.5}$ and observed PM$_{2.5}$ in 2014.

(8) Please introduce about the data source of RH and visibility in section 2.2. Generally the locations of the meteorological stations are not the same with those of the air quality stations. Did the authors use the nearest matching to pair the data? If so, what is the mean distance between the meteorological station and air quality station?

**Response:**

The visibility data we used is obtained from Global Summary of the Day (GSOD) database from National Environmental Satellite, Data, and Information Service (NESDIS) of the US Department of Commerce. Besides visibility, GSOD also provides daily average temperature and dew point, sea level pressure, wind speed and other meteorological elements and records of weather phenomena such as fog, rain and snow

(http://www.ncdc.noaa.gov/cgi-bin/res40.pl). The GSOD data undergo extensive automated quality control by the Air Weather Service, and over 400 algorithms are applied automatically to correctly 'decode' the synoptic data, and to eliminate many of the random and systematic errors found in the original data. Data are generally available from 1929 to the present.

The RH used in this study is derived from dew point temperature and local air temperature following the approach proposed by Lawrence (2005). Therefore, RH and visibility data come from the same location.

We have added more information about the data source of visibility and RH in our revised manuscript as follows:

"Winter visibility data in 1973–2019 are obtained from Global Summary of Day (GSOD) provided by the National Climatic 50 Data Center (NCDC) (https://gis.ncdc.noaa.gov/maps/ncei/cdo/daily, last access: 10 March 2022). The relative humidity (RH) is derived from dew point temperature and air temperature of GSOD following the approach proposed by Lawrence (2005)."

(9) Line 120, combined with other studies and this work, we understand that the emission is the major factor that influences the $PM_{2.5}$ trend when compared to the meteorological variables. However, in Chen et al. (2019), the meteorological factors can still account for 21% of the contribution, which is much larger than the values reported by the authors in Line 117. I do not think this is an 'agreement'.

**Response:**

Sorry for the confusion! You are right that at this stage of the paper (Lines 111–125), "we understand that the emission is the major factor that influences the $PM_{2.5}$ trend when compared to the meteorological variables". Hence any number that shows (emission >> meteorology) is considered an agreement, so is the 21% meteorology because it is much less than the 79% emission. Nevertheless, you are quite right about

"the meteorological factors can still account for 21% of the contribution, which is much larger than the values reported by the authors in Line 117". We will clarify this point in the revised manuscript.

(10) Lines 162-164, based on the analysis performed by the authors, if there exists any method that can compensate the shortcomings of the MLR and prognostic model?

**Response:**

Very important question!

As a start we believe that the alternative interpretation of MLR results proposed in Section 3.3: "The correlation coefficient should be interpreted as the maximum contribution of an independent variable to the dependent variable and the residual should be interpreted as the minimum contribution of all other independent variables" can help compensate some shortcomings of the MLR.

In regard to prognostic models, we are quite optimistic because some innovative studies have already appeared. For instance, Dang and Liao (2019) made a 33-year (1985–2017) model simulation study of severe winter haze days in BTH (purple line in Figure R4). There is an excellent agreement between the purple line and $PM_{2.5}$ concentrations observed by the US Embassy in Beijing (blue line, 2009–2018). The agreement with $PM_{2.5}$ concentrations observed by CNEMC in BTH (red line, 2013–2018) is also very good. For the entire period of 1985–2017, there are moderate mismatches near 1997–2002 and 2010 between the purple line (Dang and Liao, 2019) and green line (Li et al., 2021), but still has an acceptable overall correlation coefficient of 0.4. As cited in lines 201–202 of our paper, Dang and Liao (2019) "found that meteorology contributed significantly more than emissions to the linear trend", which is consistent with the result of our study.

[Figure]

Figure R4. Temporal variations of winter inversed PM$_{2.5}$ concentrations in BTH of this study (black, 1985–2018), simulated PM$_{2.5}$ concentrations in Beijing by Dang and Liao (2019) (purple, 1985–2017), PM$_{2.5}$ concentrations observed by the US Embassy in Beijing (blue, 2009–2018) and those observed by CNEMC in BTH (red, 2013–2018).

(11) Lines 180-185, whether this means that previous studies that focused the ASI harbor relatively large uncertainty?

**Response:**

This part of the analysis mainly emphasizes that the MLR results are highly sensitive to the length of study time. Any short term MLR study, including those involving ASI, can harbor large uncertainty.

(12) Lines 166-169, I suggest the authors to provide some theoretical foundations to support this interpretation.

**Response:**

Thank you for a highly significant suggestion. In Section 3.2 we tried but could not come up with a sound theoretical foundation for our alternative interpretation. We will continue to try in future studies.

**References**

Dang, R. and Liao, H.: Severe winter haze days in the Beijing-Tianjin-Hebei region from 1985 to 2017 and the roles of anthropogenic emissions and meteorology, Atmos. Chem. Phys., 19(16), 10801–10816, https://doi.org/10.5194/acp-19-10801-2019, 2019.

Huang, Z., Zhong, Z., Sha, Q., Xu, Y., Zhang, Z., Wu, L., Wang, Y., Zhang, L., Cui, X., Tang, M. S., Shi, B., Zheng, C., Li, Z., Hu, M., Bi, L., Zheng, J. and Yan, M.: An updated model-ready emission inventory for Guangdong Province by incorporating big data and mapping onto multiple chemical mechanisms, Sci. Total Environ., 769, 144535, https://doi.org/10.1016/j.scitotenv.2020.144535, 2021.

Lawrence, M. G.: The relationship between relative humidity and the dewpoint temperature in moist air: A simple conversion and applications, Bull. Am. Meteorol. Soc., 86(2), 225–233, https://doi.org/10.1175/BAMS-86-2-225, 2005.

Li, H., Yang, Y., Wang, H., Li, B., Wang, P., Li, J. and Liao, H.: Constructing a spatiotemporally coherent long-term $PM_{2.5}$ concentration dataset over China during 1980–2019 using a machine learning approach, Sci. Total Environ., 765, 144263, https://doi.org/10.1016/j.scitotenv.2020.144263, 2021.

Zhong, Z., Zheng, J., Zhu, M., Huang, Z., Zhang, Z., Jia, G., Wang, X., Bian, Y., Wang, Y. and Li, N.: Recent developments of anthropogenic air pollutant emission inventories in Guangdong province, China, Sci. Total Environ., 627, 1080–1092, https://doi.org/10.1016/j.scitotenv.2018.01.268, 2018.

---

## Community Comment (CC3)

Dear Editor,

We appreciate the prompt reviews and would like to thank the reviewer for insightful comments and suggestions on our manuscript entitled "Contributions of meteorology and anthropogenic emissions to the trends in winter $PM_{2.5}$ in eastern China 2013–2018" (MS No.: acp-2022-304). We have carefully considered all comments and suggestions. Listed below are our point-by-point responses to all comments and suggestions of this reviewer (Reviewer's points in black, our responses in blue).

**Anonymous Referee #2**

This is a very interesting paper analyzing the causes of $PM_{2.5}$ trends observed in eastern China. The research topic is highly important from air pollution control point of view and, although this topic has been studied quite intensively during the recent years, this paper manages to provide new insight into it. The paper is clearly organized and relatively well written. I could not find any scientific errors, even though I do feel being an expert on trend analysis. I have a few minor issues to be considered before accepting this paper for publication:

**Response:**

We appreciate the encouraging comments, particularly from an expert on trends.

**Specific comments**

(1) Please explain how ASI is defined. Based on Figure 1 it seems to be dimensionless variable but this has not been explained anywhere. This is particularly important because related to Arctic sea ice, most often the concept "Arctic sea ice area" is used in a scientific literature.

**Response:**

We adopted the Arctic Sea Ice index (ASI) suggested by Wang et al. (2015), i.e. the area-averaged sea ice fraction in the region of north 45$^o$N. Its dimension is fraction.

ASI was calculated from the Hadley Centre (HadISST1: Hadley Centre Sea Ice and Sea Surface Temperature data set, https://www.metoffice.gov.uk/hadobs/hadisst) with $1° × 1°$ resolution for 1870–2022 (Rayner et al., 2003).

(2) Similarly, please explain explicitly what is meant by "emissions" appearing in Figures 1 to 5. Are they simply primary PM emissions taken from the emission inventory, or do they also include precursors that form secondary aerosol matter?

**Response:**

The 'emissions' are composed of $PM_{10}$, $PM_{2.5}$, $SO_2$, $NH_3$, NOx, black carbon, and organic carbon in three sets of emission inventories (PKU inventory, MEIC inventory and PRD-EI inventory). These emission inventories only conclude primary emissions, precursors forming secondary aerosols are not taken into consideration. Data and calculation methods for emissions are presented in Section 2.1.

As an example, Figure R1 shows the temporal variation of three emission inventories in PRD. They show generally consistent variations during overlapping periods.

[Figure]

Figure R1. PKU emission inventory for winter 1985–2012, MEIC emission inventory for winter 2010–2016 and PRD-EI emission inventory for winter 2006–2018 for PRD. The raw data are normalized to the difference of the maximum value and minimum value.

(3) Describing what was done or observed, is usually written in a past tense. Please check out this throughout section 2. Past tense is also preferred in the following places: ··· were crucial (line 35), ··· was the most (line 37), ··· carried out (lines 111 and 128).

**Response:**

Thanks. We have checked and corrected the tense problems according to your suggestion in the revised manuscript.

(4) Section 3.2. Please reformulate the title of this section (e.g. "Comparing the MLR results ···). Starting the section by referring to "the second question" is not a good practice, as the reader need to find out from the earlier text what is this question. I would recommend repeating this section e.g. by writing "The answer to the question whether Table 1 or 2 is correct is that neither of them is correct, for the following ···"

**Response:**

Thank you for the suggestions. We have reformulated the title of Section 3.2 to be "Comparing the MLR results to mechanistic models". We also revised the beginning statement of Section 3.2 as you suggested "The answer to the question whether Table 1 or 2 is correct is that neither of them is correct, for the following reasons"

**References:**

Rayner, N. A., Parker, D. E., Horton, E. B., Folland, C. K., Alexander, L. V., Rowell, D. P., Kent, E. C. and Kaplan, A.: Global analyses of sea surface temperature, sea ice, and night marine air temperature since the late nineteenth century, J. Geophys. Res. Atmos., 108(14), https://10.1029/2002jd002670, 2003.

Wang, H. J., Chen, H. P., and Liu, J.: Arctic Sea ice decline intensified haze pollution in Eastern China, Atmos. Oceanic. Sci. Lett., 8(1), 1–9, https://doi.org/10.3878/AOSL20140081, 2015.

---

## Community Comment (CC4)

Dear Editor,

We would like to make an additional response to specific comment #12 of referee #3 on our manuscript entitled "Contributions of meteorology and anthropogenic emissions to the trends in winter PM$_{2.5}$ in eastern China 2013–2018" (MS No.: acp-2022-304). Reviewer's points in black, our responses in blue.

**Anonymous Referee #3**

(12) Lines 166-169, I suggest the authors to provide some theoretical foundations to support this interpretation.

**Response:**

In the following we present a simplistic idea about a possible theoretical foundation to support our alternative interpretation of "the maximum possible contribution of the independent variable to the dependent variable". As an example, Figure R1 below depicts an MLR analysis of the contribution of emission to the linear trend of PM$_{2.5}$ in BTH. It can be seen in Figure R1 that the MLR analysis is, in effect, performing the best-fit between the red line (emission) and the black line (observed PM$_{2.5}$). In other words, the best-fit enables the red line to attain the "maximum possible contribution" to the variability (including the linear trend) of the black line, where the "maximum" is established because all factors-other-than-emission that may contribute to the variability are excluded in the best-fit process. We propose the argument above as a possible theoretical foundation to support our alternative interpretation.

[Figure]

Figure R1. Results of MLR-EMIS analysis for 2013–2018 in BTH. Temporal variations of observed winter $PM_{2.5}$ concentration are shown in black, contributions of anthropogenic emissions to the $PM_{2.5}$ trend are shown in red, and the residual is shown in blue. Values inset in each panel are the ordinary linear regression trends, with 95% confidence intervals obtained by the student's t test.

---

## Author Response (AR1)

Dear Editor,

We appreciate the prompt reviews and would like to thank the three reviewers for insightful comments and suggestions on our manuscript entitled "Contributions of meteorology and anthropogenic emissions to the trends in winter $PM_{2.5}$ in eastern China 2013–2018" (MS No.: acp-2022-304). We have carefully considered all comments and suggestions. Listed below are our point-by-point responses to all comments and suggestions of this reviewer (Reviewer's points in black, our responses in blue).

**Anonymous Referee #1**

This paper presents a MLR statistical attribution of the 1985-2018 $PM_{2.5}$ trends in three megacity clusters in China, using visibility data as proxy for pre-2013 $PM_{2.5}$ data. It finds a large meteorological (non-emission) contribution to the trend, and argues that previous MLR analyses of the 2013-2018 trend using the actual $PM_{2.5}$ data starting in 2013 and attributing the trend to emissions are not robust. The paper makes some good points about the difficulty of sorting out meteorological effects when interpreting short (post-2013) trends. However, I believe that it may be (1) flawed in its reconstruction of the 1985-2018 $PM_{2.5}$ record which is the basis for most of the argumentation, (2) mistaken in claiming that attribution of recent $PM_{2.5}$ trends to emissions is not based on mechanistic knowledge, and (3) annoying in belaboring trivial statistical points that are well known to any trained scientist. I don't think that this paper is publishable in ACP in current form.

**Response:**

The reviewer made three general criticisms. Our responses are listed below one by one.

(1) Our reconstruction of the 1985–2018 $PM_{2.5}$ record is based on a method that converts observed visibility data to the concentrations of $PM_{2.5}$. This method has been

shown in many previous studies to be credible (Shen et al., 2016; Liu et al., 2017; Gui et al., 2020; Li et al., 2020; Li et al., 2021). The first two references are already cited in our paper. Given the serious concern of this reviewer, in Figure R1 below we compare the winter $PM_{2.5}$ record derived in our study (black line) to winter haze days derived from observed visibility data in Beijing by Li et al. (2021) (green line). The winter $PM_{2.5}$ concentrations are expected to be well correlated with the number of winter haze days. Indeed the correlation coefficient between the green line and black line is quite high at 0.7. Also shown in Figure 1 are $PM_{2.5}$ concentrations observed by the US Embassy in Beijing (blue line, 2009–2018) and those observed by CNEMC in BTH (red line, 2013–2018). The crucial bulge-2013 shows up consistently in all data sets. The correlation coefficients between our $PM_{2.5}$ values and those of the US Embassy and CNEMC are also very high at 0.6 and 0.9, respectively. These high correlation coefficients suggest that our reconstruction of the 1985–2018 $PM_{2.5}$ record from observed visibility data is credible.

[Figure]

Figure R1. Temporal variations of winter inversed $PM_{2.5}$ concentrations in BTH of this study (black, 1985–2018), simulated $PM_{2.5}$ concentrations in Beijing by Dang and Liao (2019) (purple, 1985–2017), $PM_{2.5}$ concentrations observed by the US Embassy in Beijing (blue, 2009–2018) and those observed by CNEMC in BTH (red, 2013–2018).

(2) In our paper we didn't state "that attribution of recent $PM_{2.5}$ trends to emissions is not based on mechanistic knowledge". What we did state was that *quantitative*

*attribution* of recent PM$_{2.5}$ trends to emissions is not based on *realistic/credible mechanistic models*. There is a significant difference between mechanistic knowledge and realistic/credible mechanistic models. Only realistic/credible mechanistic models have the capability of making quantitative attribution of recent PM$_{2.5}$ trends. However, it is extremely formidable to make a multi-year realistic/credible simulation of the winter mean PM$_{2.5}$ in the megacity clusters in China. In our opinion, the most realistic multi-year mechanistic model simulation is the study by Dang and Liao (2019), who made a 33-year (1985–2017) model simulation study of severe winter haze days in BTH (purple line in Figure R1). There is an excellent agreement between the purple line and PM$_{2.5}$ concentrations observed by the US Embassy in Beijing (blue line, 2009–2018). The agreement with PM$_{2.5}$ concentrations observed by CNEMC in BTH (red line, 2013-2018) is also very good. For the entire period of 1985–2017, there are moderate mismatches near 1997–2002 and 2010 between the purple line (Dang and Liao, 2019) and green line (Li et al., 2021), but still has an acceptable overall correlation coefficient of 0.4. As cited in lines 201–202 of our paper, Dang and Liao (2019) "found that meteorology contributed significantly more than emissions to the linear trend", which is consistent with the result of our study.

(3) We accept the criticism of "belaboring trivial statistical points that are well known to any trained scientist." We will delete some of the repeated statements in the revised manuscript. We were surprised that previous MLR studies would have overlooked these trivial yet important statistical points, and tried to find an explanation. That leads us to realize the importance of the bulge-2013 (as noted by this reviewer in specific comment #1 below) and to suggest the "maximum possible contribution" as an alternative interpretation for the MLR results.

**Specific comments**

1. The 'bulge-2013' feature in Figure 1 (line 88) anchors much of the argumentation in the paper but it is very weird. It seems caused by the switch from the visibility proxy to the actual PM$_{2.5}$ data in 2013. The methods are buried in Supplementary

Material. Is this 'bulge-2013' seen in the consistent long-term satellite AOD data record? I think that the authors would have to show that it is present in the AOD data in order to have credibility.

**Response:**

We believe that Figure R1 and associated discussions above address this comment adequately. In response to the question about AOD, we compare below the winter satellite AOD data (MERRA2) in BTH to $PM_{2.5}$ and visibility (both from this study) in Figure R2. The correlation between AOD and $PM_{2.5}$ is fair (overall correlation coefficient 0.3) except some mismatches during two periods (2007–2009 and 2012–2013). As a result, only half of the bulge (2013–2018) can be seen in AOD. The reason for the mismatches is probably because surface $PM_{2.5}$ is sensitive to the height of mixed layer while AOD is not. In other words, changes of surface $PM_{2.5}$ due to changing mixing height are usually not detected in the AOD observations.

[Figure]

Figure R2. Time series of winter $PM_{2.5}$ concentration, visibility (both from our study) and AOD (from MERRA2) in BTH from 1985 to 2018.

2. What is the 'emission' in Figure 1? Of what species?

**Response:**

The 'emission' is composed of $PM_{10}$, $PM_{2.5}$, $SO_2$, $NH_3$, NOx, black carbon, and organic carbon in three sets of emission inventories (PKU inventory, MEIC inventory and PRD-EI inventory). Data and calculation methods for emissions are presented in

Section 2.1.

As an example, Figure R3 shows the temporal variation of three emission inventories in PRD. They show generally consistent variations during overlapping periods.

[Figure]

Figure R3. PKU emission inventory for winter 1985–2012, MEIC emission inventory for winter 2010–2016 and PRD-EI emission inventory for winter 2006–2018 for PRD. The raw data are normalized to the difference of the maximum value and minimum value.

3. Line 91: the Mao et al. 2019 reference which is intended to provide support for the authors' meteorological attribution of the trend is in fact grey literature involving some of the same authors.

**Response:**

Mao et al. (2019) is a peer reviewed article, not a "grey literature". The Mao et al. (2019) reference (Lines 352–353) is reproduced below:

Mao, L., Liu, R., Liao, W., Wang, X., Shao, M., Liu, S. C. and Zhang, Y.: An observation-based perspective of winter haze days in four major polluted regions of China, Natl. Sci. Rev., 6(3), 515–523, https://doi.org/10.1093/nsr/nwy118, 2019.

National Science Review is published by Oxford University Press on behalf of China

Science Publishing & Media Ltd. The current impact factor of National Science Review is over 23, which ranks it among the best international scientific journals.

4. Line 132, etc.: the mechanistic meteorological connection of ASI to $PM_{2.5}$ is not clear, and as the authors point out any meteorological variable with a suitable long-term trend would do the trick. But there is in fact a strong mechanistic argument for emissions to be related to $PM_{2.5}$ (line 154), and there is strong independent evidence that Chinese emissions have decreased over the 2013-2018 period (emission inventories, satellite data). To claim that the connection of $PM_{2.5}$ to emissions has no mechanistic support strikes me as obviously wrong. In fact the authors cite Chen et al. 2019 in demonstrating the mechanistic connection in WRF-CMAQ but argue that the analysis is flawed because it did not consider the effect of the bulge-2013 (line 158). As pointed out above, I am very suspicious of this bulge-2013.

**Response:**

We have addressed extensively the issues raised here in general comment #2 (and #1 about the bulge-2013). Moreover, in lines 261–262 we already stated "there is little doubt that anthropogenic emissions make a significant contribution to the reduction trend of $PM_{2.5}$." We were only "skeptical of *those high* contributions by emissions obtained based solely on MLR models."

5. There is a lot of trivial stuff about the non-mechanistic basis of statistical models, correlation not implying causality, more years increasing the credibility of the model, etc., that is repeated again and again and does not rise above the level of a basic course in statistics.

**Response:**

As stated in our response to general comment #3, we accept the criticism of "belaboring trivial statistical points that are well known to any trained scientist," We will delete some of the repeated statements in the revised manuscript.

**Anonymous Referee #2**

This is a very interesting paper analyzing the causes of PM$_{2.5}$ trends observed in eastern China. The research topic is highly important from air pollution control point of view and, although this topic has been studied quite intensively during the recent years, this paper manages to provide new insight into it. The paper is clearly organized and relatively well written. I could not find any scientific errors, even though I do feel being an expert on trend analysis. I have a few minor issues to be considered before accepting this paper for publication:

**Response:**

We appreciate the encouraging comments, particularly from an expert on trends.

**Specific comments**

(1) Please explain how ASI is defined. Based on Figure 1 it seems to be dimensionless variable but this has not been explained anywhere. This is particularly important because related to Arctic sea ice, most often the concept "Arctic sea ice area" is used in a scientific literature.

**Response:**

We adopted the Arctic Sea Ice index (ASI) suggested by Wang et al. (2015), i.e. the area-averaged sea ice fraction in the region of north 45$^{o}$N. Its dimension is fraction. ASI was calculated from the Hadley Centre (HadISST1: Hadley Centre Sea Ice and Sea Surface Temperature data set, https://www.metoffice.gov.uk/hadobs/hadisst) with 1$^{o}$ × 1$^{o}$ resolution for 1870–2022 (Rayner et al., 2003).

(2) Similarly, please explain explicitly what is meant by "emissions" appearing in Figures 1 to 5. Are they simply primary PM emissions taken from the emission inventory, or do they also include precursors that form secondary aerosol matter?

**Response:**

The 'emissions' are composed of $PM_{10}$, $PM_{2.5}$, $SO_2$, $NH_3$, NOx, black carbon, and organic carbon in three sets of emission inventories (PKU inventory, MEIC inventory and PRD-EI inventory). These emission inventories only conclude primary emissions, precursors forming secondary aerosols are not taken into consideration. Data and calculation methods for emissions are presented in Section 2.1.

As an example, Figure R3 shows the temporal variation of three emission inventories in PRD. They show generally consistent variations during overlapping periods.

(3) Describing what was done or observed, is usually written in a past tense. Please check out this throughout section 2. Past tense is also preferred in the following places: ⋯ were crucial (line 35), ⋯ was the most (line 37), ⋯ carried out (lines 111 and 128).

**Response:**

Thanks. We have checked and corrected the tense problems according to your suggestion in the revised manuscript.

(4) Section 3.2. Please reformulate the title of this section (e.g. "Comparing the MLR results ⋯). Starting the section by referring to "the second question" is not a good practice, as the reader need to find out from the earlier text what is this question. I would recommend repeating this section e.g. by writing "The answer to the question whether Table 1 or 2 is correct is that neither of them is correct, for the following ⋯"

**Response:**

Thank you for the suggestions. We have reformulated the title of Section 3.2 to be "Comparing the MLR results to mechanistic models". We also revised the beginning statement of Section 3.2 as you suggested "The answer to the question whether Table 1 or 2 is correct is that neither of them is correct, for the following reasons"

This work proposes a different method for the MLR analysis of PM$_{2.5}$. Based on the new interpretation and the comparison with previous studies, the MLR results among different studies were found to be more consistent. In addition, the authors also pointed out that the relationship constrained by long-term data is more reliable. Overall, this is an interesting study and it provides some useful information for other researchers when choosing MLR for air quality trend analysis. However, more explanations, especially for the methodology, are still needed.

**Response:**

We appreciate the insightful comments and suggestions. More explanations have been added in our revised manuscript, especially in the methodology section.

**Specific comments**

(1) Line 60, the resolution of the PRD emission inventory is three degrees, which is rather coarse.

**Response:**

The emission inventory of PRD (PRD-EI) is adopted from Huang et al. (2021) and Zhong et al. (2018). Although the resolution of the PRD-EI is coarser than other two inventories, we can only get the emission information of year 2018 and 2019 from PRD-EI. In addition, in our study, we mainly focus on the long-term trend and interannual variation in the annual total emissions in each region, which is independent of the resolution of emission inventories.

Figure R3 shows the temporal variation of three emission inventories. They have similar variation for the overlapping period.

(2) This work mainly focuses on the PRD, YRD and Jing-Jin-Ji regions. For YRD and Jing-Jin-Ji regions, the authors combined the MEIC and PKU emission inventories to

do the analysis. While for PRD, they combined PKU and PRD-EI to do the scaling. MEIC and PRD-EI are different emission inventories and the methods that used to derive these two emission inventories should be not consistent. Based on the literatures, the MEIC emission inventory should have already covered the PRD region, why not also using the MEIC emission inventory to analyze the PRD region?

**Response:**

The MEIC inventory does also cover the PRD region, but the time span of MEIC inventory is 2010–2017. The time span of the PRD-EI inventory and PKU inventory is 2006–2019 and 1960–2014, repetitively. Therefore, we combined PRD-EI and PKU inventories in PRD for the winters of 1985–2018.

(3) Please label the scaling factor and Ei equations.

**Response:**

Thanks, we have labeled the scaling factor and $E_i$ equations in our revised manuscript.

(4) Line 71, please use data or reference to support this assumption.

**Response:**

Thanks, we have added Figure S1 (Figure R4) in the revised Supplementary Material and revised the Line 71 statement to make it clearer as "Since the ratios of annual emission inventory in PRD to those of YRD and BTH are not expected to change significantly in one or two years (Figure S1)"

[Figure]

Figure S1 (Figure R4). Time series of emission inventory (EI) ratios in the winter of 1985–2018 for the BTH/PRD and YRD/PRD, respectively.

(5) The PRD scaling factor was calculated by the emission sum from 2006 to 2013, while the scaling factors for the other two regions were calculated by the emission sum from 2010 to 2013. Please explain why using different emission sum to derive the scaling factors.

**Response:**

We calculate the scaling factor based on the overlapping periods of two inventories. As stated in our response to your Point #2, the time spans of these three inventories are different, so we derived scaling factors for PRD from 2006 to 2013, while for BTH and YRD from 2010 to 2013.

(6) The authors applied the nonlinear exponential fitting to retrieve the long-term PM$_{2.5}$ concentration before 2013, because China began to release the air quality observation data since 2013 and it is unlikely to acquire long-term observation data in this nation before 2013. However, based on the figures in the supplemental material, some of the fittings are not acceptable for further analysis, such as BTH-RH (40, 60) and YRD-RH (90, 100). The authors need to analyze and discuss whether such errors can influence their conclusion.

**Response:**

We believe that our PM$_{2.5}$ retrievals from nonlinear exponential fitting are acceptable for further analysis and do not influence our conclusion for the following reasons: (1) Yes, the correlation coefficient (R) of 0.56 for BTH-RH (40, 60) is a little low, but the fittings for other intervals with more samples are very good, so the overall fitting for BTH remains good. The sample size of YRD-RH (90, 100) is the smallest among all intervals, so its relatively small R (0.36) has negligible effect on the overall fitting for YRD. (2) We compared the retrieved PM$_{2.5}$ concentrations with the observed PM$_{2.5}$ concentration in BTH and YRD (Figure R5), and found that R is more than 0.87, and normalized mean bias (NMB) are 5.9% and 4.6% in BTH and YRD, respectively. These values of R and NMB suggest that the exponential fitting model is capable of reproducing the observed PM$_{2.5}$ concentrations. (3) As you suggested in comment point #7, we use the data of 2015–2019 for the fitting and the 2014 data for the verification, the R (NMB) between the fitted PM$_{2.5}$ concentrations and observed PM$_{2.5}$ concentrations is 0.77 (14.8%), 0.84 (5.5%), and 0.93 (5.1%) in BTH, YRD and PRD, respectively, suggesting that the exponential fitting models are robust. (4) R between the long-term retrieved PM$_{2.5}$ concentrations and observed visibility reached -0.9 in both BTH and YRD, reconfirming that the performance of our exponential fitting model is satisfactory (Figures S4a-b).

[Figure]

Figure R5. Temporal variation of retrieved PM$_{2.5}$ and observed PM$_{2.5}$ from 2015 to 2018.

(7) For the PM$_{2.5}$ concentration retrieval, I suggest the authors use the data of 2014-2018 for the fitting and the 2013 data for the verification, this can help to verify whether the methods implemented by the authors are reliable or not.

**Response:**

Since we did not have daily PM$_{2.5}$ concentration data for the three regions in 2013, we used observed daily visibility and PM$_{2.5}$ concentration from 2015 to 2019 to establish the exponential fitting model, as shown in Figures S1–S3. Furthermore, we use 2014 daily fitted PM$_{2.5}$ and observed PM$_{2.5}$ for verification (Figure R6). The R (NMB) between the fitted PM$_{2.5}$ concentration and observed PM$_{2.5}$ concentration is 0.77 (14.8%), 0.84 (5.5%), and 0.93 (5.1%) in BTH, YRD and PRD, respectively, verifying that the exponential fitting models are reliable.

[Figure]

Figure R6. Temporal variation of retrieved PM$_{2.5}$ and observed PM$_{2.5}$ in 2014.

(8) Please introduce about the data source of RH and visibility in section 2.2. Generally the locations of the meteorological stations are not the same with those of the air quality stations. Did the authors use the nearest matching to pair the data? If so, what is the mean distance between the meteorological station and air quality station?

**Response:**

The visibility data we used is obtained from Global Summary of the Day (GSOD) database from National Environmental Satellite, Data, and Information Service (NESDIS) of the US Department of Commerce. Besides visibility, GSOD also provides daily average temperature and dew point, sea level pressure, wind speed and other meteorological elements and records of weather phenomena such as fog, rain and snow (http://www.ncdc.noaa.gov/cgi-bin/res40.pl). The GSOD data undergo extensive automated quality control by the Air Weather Service, and over 400

algorithms are applied automatically to correctly 'decode' the synoptic data, and to eliminate many of the random and systematic errors found in the original data. Data are generally available from 1929 to the present.

The RH used in this study is derived from dew point temperature and local air temperature following the approach proposed by Lawrence (2005). Therefore, RH and visibility data come from the same location.

We have added more information about the data source of visibility and RH in our revised manuscript as follows:

"Winter visibility data in 1973–2019 are obtained from Global Summary of Day (GSOD) provided by the National Climatic Data Center (NCDC) (https://www.ncei.noaa.gov/maps/daily/, last access: 10 March 2022). The relative humidity (RH) is derived from dew point temperature and air temperature of GSOD following the approach proposed by Lawrence (2005)."

(9) Line 120, combined with other studies and this work, we understand that the emission is the major factor that influences the $PM_{2.5}$ trend when compared to the meteorological variables. However, in Chen et al. (2019), the meteorological factors can still account for 21% of the contribution, which is much larger than the values reported by the authors in Line 117. I do not think this is an 'agreement'.

**Response:**

Sorry for the confusion! You are right that at this stage of the paper (Lines 111–125), "we understand that the emission is the major factor that influences the $PM_{2.5}$ trend when compared to the meteorological variables". Hence any number that shows (emission >> meteorology) is considered an agreement, so is the 21% meteorology because it is much less than the 79% emission. Nevertheless, you are quite right about "the meteorological factors can still account for 21% of the contribution, which is much larger than the values reported by the authors in Line 117". We will clarify this

point in the revised manuscript.

(10) Lines 162-164, based on the analysis performed by the authors, if there exists any method that can compensate the shortcomings of the MLR and prognostic model?

**Response:**

Very important question!

As a start we believe that the alternative interpretation of MLR results proposed in Section 3.3: "The correlation coefficient should be interpreted as the maximum contribution of an independent variable to the dependent variable and the residual should be interpreted as the minimum contribution of all other independent variables" can help compensate some shortcomings of the MLR.

In regard to prognostic models, we are quite optimistic because some innovative studies have already appeared. For instance, Dang and Liao (2019) made a 33-year (1985–2017) model simulation study of severe winter haze days in BTH (purple line in Figure R7). There is an excellent agreement between the purple line and $PM_{2.5}$ concentrations observed by the US Embassy in Beijing (blue line, 2009–2018). The agreement with $PM_{2.5}$ concentrations observed by CNEMC in BTH (red line, 2013–2018) is also very good. For the entire period of 1985–2017, there are moderate mismatches near 1997–2002 and 2010 between the purple line (Dang and Liao, 2019) and green line (Li et al., 2021), but still has an acceptable overall correlation coefficient of 0.4. As cited in lines 201–202 of our paper, Dang and Liao (2019) "found that meteorology contributed significantly more than emissions to the linear trend", which is consistent with the result of our study.

[Figure]

Figure R7. Temporal variations of winter inversed $PM_{2.5}$ concentrations in BTH of this study (black, 1985–2018), simulated $PM_{2.5}$ concentrations in Beijing by Dang and Liao (2019) (purple, 1985–2017), $PM_{2.5}$ concentrations observed by the US Embassy in Beijing (blue, 2009–2018) and those observed by CNEMC in BTH (red, 2013–2018).

(11) Lines 180-185, whether this means that previous studies that focused the ASI harbor relatively large uncertainty?

**Response:**

This part of the analysis mainly emphasizes that the MLR results are highly sensitive to the length of study time. Any short term MLR study, including those involving ASI, can harbor large uncertainty.

(12) Lines 166-169, I suggest the authors to provide some theoretical foundations to support this interpretation.

**Response:**

In the following we present a simplistic idea about a possible theoretical foundation to support our alternative interpretation of "the maximum possible contribution of the independent variable to the dependent variable". As an example, Figure R8 below depicts an MLR analysis of the contribution of emission to the linear trend of $PM_{2.5}$ in

BTH. It can be seen in Figure R8 that the MLR analysis is, in effect, performing the best-fit between the red line (emission) and the black line (observed PM$_{2.5}$). In other words, the best-fit enables the red line to attain the "maximum possible contribution" to the variability (including the linear trend) of the black line, where the "maximum" is established because all factors-other-than-emission that may contribute to the variability are excluded in the best-fit process. We propose the argument above as a possible theoretical foundation to support our alternative interpretation.

[Figure]

Figure R8. Results of MLR-EMIS analysis for 2013–2018 in BTH. Temporal variations of observed winter PM$_{2.5}$ concentration are shown in black, contributions of anthropogenic emissions to the PM$_{2.5}$ trend are shown in red, and the residual is shown in blue. Values inset in each panel are the ordinary linear regression trends, with 95% confidence intervals obtained by the student's t test.

**References**

Dang, R. and Liao, H.: Severe winter haze days in the Beijing-Tianjin-Hebei region from 1985 to 2017 and the roles of anthropogenic emissions and meteorology, Atmos. Chem. Phys., 19(16), 10801–10816, https://doi.org/10.5194/acp-19-10801-2019, 2019.

Gui, K., Che, H., Zeng, Z., Wang, Y., Zhai, S., Wang, Z., Luo, M., Zhang, L., Liao, T., Zhao, H., Li, L., Zheng, Y. and Zhang, X.: Construction of a virtual $PM_{2.5}$ observation network in China based on high-density surface meteorological observations using the Extreme Gradient Boosting model, Environ. Int., 141, 105801, https://doi.org/10.1016/j.envint.2020.105801, 2020.

Huang, Z., Zhong, Z., Sha, Q., Xu, Y., Zhang, Z., Wu, L., Wang, Y., Zhang, L., Cui, X., Tang, M. S., Shi, B., Zheng, C., Li, Z., Hu, M., Bi, L., Zheng, J. and Yan, M.: An updated model-ready emission inventory for Guangdong Province by incorporating big data and mapping onto multiple chemical mechanisms, Sci. Total Environ., 769, 144535, https://doi.org/10.1016/j.scitotenv.2020.144535, 2021.

Lawrence, M. G.: The relationship between relative humidity and the dewpoint temperature in moist air: A simple conversion and applications, Bull. Am. Meteorol. Soc., 86(2), 225–233, https://doi.org/10.1175/BAMS-86-2-225, 2005.

Li, H., Yang, Y., Wang, H., Li, B., Wang, P., Li, J. and Liao, H.: Constructing a spatiotemporally coherent long-term $PM_{2.5}$ concentration dataset over China during 1980–2019 using a machine learning approach, Sci. Total Environ., 765, 144263, https://doi.org/10.1016/j.scitotenv.2020.144263, 2021.

Li, S., Chen, L., Huang, G., Lin, J., Yan, Y., Ni, R., Huo, Y., Wang, J., Liu, M., Weng, H., Wang, Y. and Wang, Z.: Retrieval of surface $PM_{2.5}$ mass concentrations over North China using visibility measurements and GEOS-Chem simulations, Atmos. Environ., 222, 117–121, https://doi.org/10.1016/j.atmosenv.2019.117121, 2020.

Liu, M., Bi, J. and Ma, Z.: Visibility-based $PM_{2.5}$ concentrations in China: 1957–1964 and 1973–2014, Environ. Sci. Technol., 51(22), 13161–13169, https://doi.org/10.1021/acs.est.7b03468, 2017.

Mao, L., Liu, R., Liao, W., Wang, X., Shao, M., Liu, S. C. and Zhang, Y.: An observation-based perspective of winter haze days in four major polluted regions of China, Natl. Sci. Rev., 6(3), 515–523, https://doi.org/10.1093/nsr/nwy118, 2019.

Rayner, N. A., Parker, D. E., Horton, E. B., Folland, C. K., Alexander, L. V., Rowell, D. P., Kent, E. C. and Kaplan, A.: Global analyses of sea surface temperature, sea ice, and night marine air temperature since the late nineteenth century, J. Geophys. Res. Atmos., 108(14), https://10.1029/2002jd002670, 2003.

Shen, Z., Cao, J., Zhang, L., Zhang, Q., Huang, R. J., Liu, S., Zhao, Z., Zhu, C., Lei, Y., Xu, H. and Zheng, C.: Retrieving historical ambient $PM_{2.5}$ concentrations using existing visibility measurements in Xi'an, Northwest China, Atmos. Environ., 126, 15–20, https://doi.org/10.1016/j.atmosenv.2015.11.040, 2016.

Wang, H. J., Chen, H. P., and Liu, J.: Arctic Sea ice decline intensified haze pollution in Eastern China, Atmos. Oceanic. Sci. Lett., 8(1), 1–9, https://doi.org/10.3878/AOSL20140081, 2015.

Zhong, Z., Zheng, J., Zhu, M., Huang, Z., Zhang, Z., Jia, G., Wang, X., Bian, Y., Wang, Y. and Li, N.: Recent developments of anthropogenic air pollutant emission inventories in Guangdong province, China, Sci. Total Environ., 627, 1080–1092, https://doi.org/10.1016/j.scitotenv.2018.01.268, 2018.

---

## Author Response (AR2)

Dear Editor,

We appreciate the prompt reviews and would like to thank the reviewers for insightful comments and suggestions on our manuscript entitled "Contributions of meteorology and anthropogenic emissions to the trends in winter PM$_{2.5}$ in eastern China 2013–2018" (MS No.: acp-2022-304). We have carefully considered all comments and suggestions. Listed below are our point-by-point responses to all comments and suggestions of Referee #3 (Reviewer's points in black, our responses in blue).

**Anonymous Referee #3**

The authors have generally answered my questions, but before accepting this manuscript, I still have two minor concerns:

1) Following my previous question 5, as the authors mentioned, the PKU-FUEL emission inventory is from 1960 to 2014. Why not include the year 2014 to do the scaling? The MEIC emission inventory is from 2010-2017. Instead of using the 2016 BTH-PRD and YRD-PRD ratios, why not directly use the formulas in Section 2.1 to do the scaling for the year 2017?

**Response:**

Our winter months are defined as December of the current year, and January and February of the following year. Therefore, we used January and February of 2014 in the PKU-FUEL emission inventory, combined with December of 2013 to make the scaling for winter 2013. For winter 2017 MEIC scaling, we needed the MEIC emission inventory for December 2017, and January and February 2018. Since we did not have the 2018 MEIC emission inventory, the 2016 BTH-PRD and YRD-PRD ratios were used to obtain the 2017 and 2018 winter emissions in BTH and YRD.

Thanks to this comment. We have added one statement in the revised manuscript "It should be noted that the winter of a specific year in this study includes December of this year and January and February of the following year." at the end of Section 2.1 to

clear possible confusion.

2) Following my previous question 7, though the statistical performance is acceptable (but indeed, R=0.77 is not perfect), a significant discrepancy can still be found before October 2014 in Figure R3. Please discuss in the manuscript whether such an error (not only for this year) can influence your conclusion.

**Response:**

In the verification of BTH in 2014, large differences between the retrieved $PM_{2.5}$ and the observed $PM_{2.5}$ mainly occurred in July-September (Figure RR1 below, which was Figure R3 in the first-round response and referred to in the comment above). In this study, our focus is on the winter season (December-February). As can be seen from Figure RR2, the retrieved $PM_{2.5}$ reproduces well the variations and trends in the observed $PM_{2.5}$ in December 2014, with a correlation coefficient (R) of 0.76, and a small normalized mean bias (NMB) of 5.2%, indicating that such an error would have negligible effect on our conclusions. Furthermore, in our original manuscript, we have compared the results derived from observed $PM_{2.5}$ and retrieved $PM_{2.5}$ (the first four rows in Table 5). The self-consistent results reconfirm the discrepancy between observed and retrieved $PM_{2.5}$ does not influence our conclusion.

[Figure]

Figure RR1 Temporal variation of retrieved PM$_{2.5}$ and observed PM$_{2.5}$ in 2014.

[Figure]

Figure RR2 Temporal variation of retrieved PM$_{2.5}$ and observed PM$_{2.5}$ in BTH in December 2014.